

# Advances in characterization of black carbon particles and their associated coatings using the soot particle aerosol mass spectrometer in Singapore, a complex city environment

5 Mutian Ma[1*], Laura-Hélèna Rivellini[2], Yichen Zong[3,4], Markus Kraft[4,5], Liya E. Yu[1,2], Alex King Yin Lee[6]

[1]Department of Civil and Environmental Engineering, National University of Singapore, Singapore

10 [2]NUS Environmental Research Institute, National University of Singapore, Singapore

[3]Department of Mechanical Engineering, National University of Singapore, Singapore

[4]Cambridge Centre of Advance Research and Education in Singapore, Singapore

[5]Department of Chemical Engineering and Biotechnology, University of Cambridge, United Kingdom'

[6]Air Quality Processes Research Section, Environment and Climate Change Canada, ON, Toronto, Canada

*now at Cambridge Centre of Advance Research and Education in Singapore, Singapore

Correspondence to: Alex King Yin, Lee (alex.lee@ec.gc.ca)



## Abstract

Atmospheric black carbon can act as a short-lived climate forcer and carrier of toxics. This work aims to utilize aerosol compositions detected by a soot-particle aerosol mass spectrometer to advance our understanding of emission and atmospheric processing of refractory BC (rBC) in Singapore. Positive matrix factorization analysis of rBC and organic aerosols (OA)

($PMF_{base}$) identified two local traffic factors with large differences in rBC content and coating thickness, and two secondary OA (SOA) factors impacted by local chemistry and/or regional transport (less-oxidized oxygenated OA (LO-OOA) and more-oxidized OA (MO-OOA)). Including metals in the PMF ($PMF_{metal}$) improved the quality of source apportionment significantly. An industrial and shipping influenced OA separated from traffic emissions was strongly associated with heavy metals (e.g., $V^+$ and $Ni^+$) that might pose higher potential risks to human health. Two biomass burning-influenced OA (BBOA) factors with

different degree of oxygenation were also identified. Although the aged-BBOA component was highly oxidized, its strong association with $K_3SO_4^+$ differentiated itself from other background SOA. Integration of both metals and inorganic aerosols (IA) into the PMF ($PMF_{all}$) was found to provide further insight into the origin of SOA coatings and their chemical processing. $PMF_{all}$ identified an additional aged-BBOA component that was associated with nighttime IA and organo-nitrate formation. Furthermore, $PMF_{all}$ revealed concurrent LO-OOA and nitrate formation during daytime, whereas photochemical production

of MO-OOA was linked to acidic sulfate formation, indicating the importance of investigating interaction between SOA and IA formation and their mixing state in complex city environments.





## 1. Introduction

Atmospheric black carbon (BC) particles generated by incomplete combustion of fossil fuel and biomass can have significant impacts on climate and human health. BC particles play a crucial role in radiative forcing, primarily through their ability to
absorb sunlight (Bond et al., 2013). When deposited on snow and ice surfaces, BC particles reduce their albedo, accelerating melting and amplifying the effects of climate change (Bond et al., 2013). BC particles are always co-emitted with other particulate matter (PM) and atmospheric processing can lead to formation of secondary PM that can subsequently condense onto BC particles surface (Bond et al., 2013; China et al., 2013; Lee et al., 2017; Collier et al., 2018). Coating formation on BC particles can modify their morphology, light absorbing properties, and chemical interaction with other atmospheric species,
and hence their environmental fate and impacts (Kuwata et al., 2009; China et al., 2013; Fierce et al., 2016; Peng et al., 2016; Wu et al., 2018; Cappa et al., 2019). Furthermore, BC particles can co-emit and internally mix with toxic contaminants such as heavy metals and polycyclic hydrocarbon carbon (PAH) from combustion processes (Corbin et al., 2018; Wang et al., 2021) and thus pose significant risks to human and ecosystem health.

Both primary and secondary organic aerosols (POA and SOA) have been found to be the major components of BC coatings based on soot particle aerosol mass spectrometer (SP-AMS) measurements (Willis et al., 2016; Lee et al., 2017; Collier et al., 2018; Wang et al., 2020; Bibi et al., 2021; Cao et al., 2022; Farley et al., 2024) as the laser vaporization mode of SP-AMS allows detection of refractory BC (rBC, an operationally defined BC detected by SP-AMS) and their coating materials exclusively (Onasch et al., 2012). Although previous studies report different fullerene rBC fragments characteristics to
differentiate burning sources (Onasch et al., 2015; Wang et al., 2016; Bibi et al., 2021), the interpretation of positive matrix factorization (PMF) solutions for distinguishing specific rBC-containing particles sources based on mass spectral characteristics of rBC and organic aerosols (OA) coating remains challenging. The mass spectral profiles of POA emitted from various fossil fuel combustion sources are generally similar to each other and largely contributed by hydrocarbon ($C_xH_y^+$) fragments without distinctive differences (Dallmann et al., 2014; Corbin et al., 2018; Carbone et al., 2019; Ma et al., 2023).
Furthermore, identifying the origins of highly oxidized/aged OA coatings are difficult as substantial atmospheric oxidative aging always diminish the mass spectral features of POA, and leads to SOA products that are mainly dominated by a single OA fragment (i.e., $CO_2^+$) as illustrated in the observational frameworks developed based on the ambient AMS measurements worldwide (Ng et al., 2010; Cubison et al., 2011).

Previous studies have shown that the laser vaporization mode of SP-AMS can detect metals in both laboratory-generated and ambient rBC particles as well as engine emissions (Dallmann et al., 2014; Carbone et al., 2015; Corbin et al., 2018; Rivellini et al., 2020; Bibi et al., 2021; Cao et al., 2022; Tehrani et al., 2023). Given that higher stability of trace metals compared to OA and the strong association of metals with combustion-related emissions, such unique measurement capability of SP-AMS




has been used to obtain additional chemical information for improving source identification of ambient rBC and OA particles and understanding emission characteristics of specific combustion processes. For examples, Bibi et al. (2021) clearly identified strong associations between different trace metals, in particularly strontium (Sr), and fireworks when multiple primary emission sources were observed by SP-AMS during the bonfire events (the Bonfire Night 2014 in Manchester, UK). Saarikoski
et al. (2016) detected 15 trace metals in bus emissions that were dominated by rBC and OA, and the relationships between the metals and rBC (e.g., Vanadium (V) and Zinc (Zn)) were different for acceleration and steady driving based on their SP-AMS observations. Corbin et al. (2018) investigated trace metals in heavy-fuel-oil combustion in a marine engine using an SP-AMS, and demonstrated that vanadium-to-nickel (V/Ni) ratios in soot were much higher than those observed in $PM_{2.5}$ regardless of the engine loads. Cao et al. (2022) observed relatively high iron (Fe) signals on rBC particles in urban area and hypothesized
that Fenton chemistry might catalyze the observed formation of more oxidized SOA. A few studies have shown that potassium (K) measured by SP-AMS and standard AMS could be used to indicate potential influences of biomass burning emissions on rBC and OA particles in different urban and rural environments (Lee et al., 2016; Wang et al., 2017; Rivellini et al., 2020; Farley et al., 2023; Vasilakopoulou et al., 2023).

Singapore is a highly developed coastal urban environment located at the end of the Malay Peninsula in Southeast Asia. Aside from typical engine emissions from local traffic, Singapore has the world's second busiest shipping port (World Shipping Council, 2024) and fifth largest oil refining capacity (Lau et al., 2021), which can be potential large anthropogenic combustion sources of air pollutants. Furthermore, together with atmospheric processing during transport, air quality and concentrations of atmospheric PM in Singapore can be significantly influenced by transboundary haze due to regional/transported biomass
burning emissions in Southeast Asia (Heil and Goldammer, 2001; Budisulistiorini et al., 2018; Kuwata, 2024). However, the impact of regional biomass burning emissions on background rBC and SOA levels remains largely unknown. Given the unique capability of SP-AMS to detect rBC particles and their associated coatings, including OA, inorganic aerosol (IA), and metals, simultaneously, this work aims to explore how the inclusion of increasingly comprehensive chemical information in PMF analysis of SP-AMS measurements advances our understanding of emissions and atmospheric processing of rBC-containing
particles in Singapore. To this end, a few PMF scenarios were considered which iteratively incorporated metals and IA into the analysis. Our step-by-step PMF interpretation approach demonstrates the importance of integrating IA and/or metals in the PMF input for improving source apportionment of rBC and OA at our sampling locations and provides insight into our understanding of secondary processing of rBC-containing particles, especially for those originally from biomass burning.

## 2.   Methodology

### 2.1 Sampling site and collocated measurements

The measurements were conducted from January 28th to March 17th, 2022, at E2 building on the campus of National University of Singapore (1◦18′ N, 103◦46′ E; 67 m above sea level). Fig. 1 shows the sampling site location and details of surrounding





environments. The air quality at the site can be significantly influenced by anthropogenic emissions as reported by our previous work (Rivellini et al., 2020). The measurement site was located at the south of the AYE highway and at the north-east of multiple shipping ports and industrial regions with refineries and petrochemical plants. A SP-AMS (Aerodyne Research) was deployed to measure chemical composition of submicron rBC-containing particles (see Section 2.2 for the details of SP-AMS

5 measurements). Co-located BC mass concentration was measured using a seven-wavelength aethalometer (AE33, Magee Scientific) (Hansen et al., 1984; Drinovec et al., 2015) and non-refractory $PM_1$ (NR-$PM_1$) was measured by a Time-of-Flight aerosol chemical speciation monitor (ToF-ACSM, Aerodyne Research) (Fröhlich et al., 2013). The SP-AMS, ToF-ACSM, and aethalometer shared a Teflon-coated cyclone inlet with the cutoff size of 2.5 μm (model 2000-30ED, URG) and the aerosol stream was dried by a Nafion™ dryer system (Aerodyne Research). Gas-phase species ($NO_x$ and $O_3$) were measured by a low-

10 cost sensor system (AQMesh air quality measurement system, Environmental Instruments Ltd., UK). Meteorological parameters were measured, including wind speed and direction (03001 RM Young Wind Sentry Set, Campbell Scientific Inc.), solar radiation (LI-COR model LI-200X, Campbell Scientific Inc.), rainfall (Hydro-logical Services CS700, Campbell Scientific Inc.), relative humidity (RH), and temperature (AQMesh air quality measurement system, Environmental Instruments Ltd., UK).

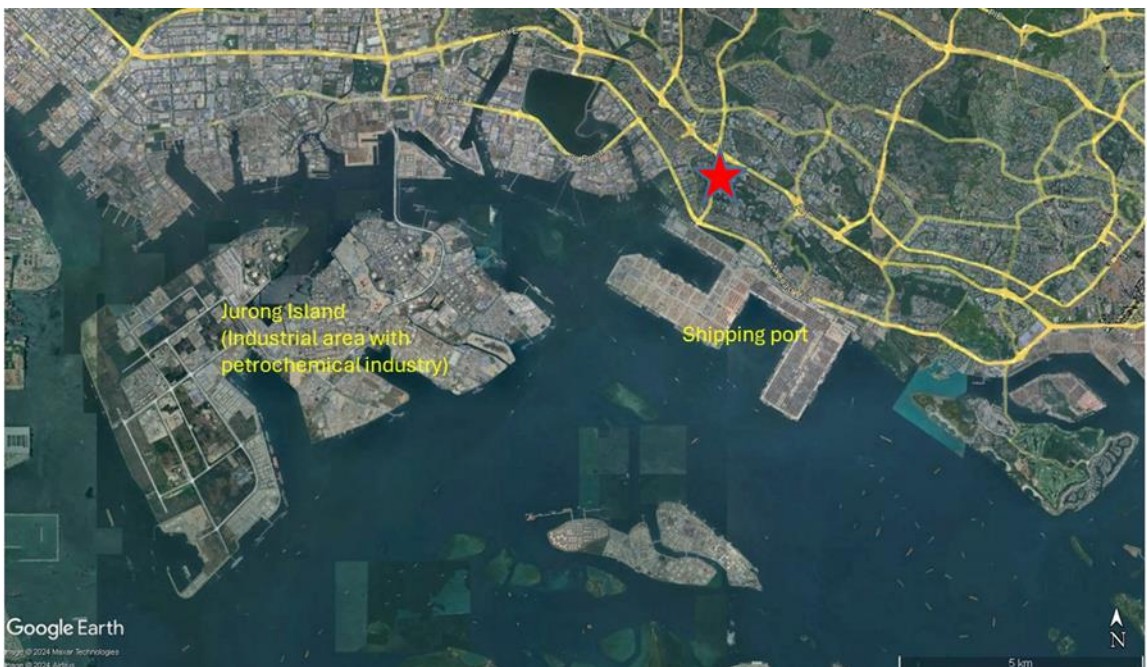

Figure 1. Sampling location (red star) and surrounding environments. The yellow lines indicate the major roadways. A large industrial area with petrochemical industries and shipping ports is located at the south of the sampling location (sources: © 20 Google Earth 2024).



## 2.2 Soot particle aerosol mass spectrometer (SP-AMS)

SP-AMS is the standard high-resolution time-of-flight aerosol mass spectrometer (HR-ToF-AMS, Aerodyne Research)
equipped with a laser vaporizer (LV). In this work, the SP-AMS was configured to detect rBC- containing particles (rBC is an
operationally defined term that refers to the BC particles detected by SP-AMS) and their associated coating materials, including
organic (Org), nitrate ($NO_3^-$), sulfate ($SO_4^{2-}$), chloride ($Cl^-$), ammonium ($NH_4^+$) and trace metals in $PM_1$ (Carbone et al., 2015;
Lee et al., 2017; Corbin et al., 2018). The resistively heated tungsten vaporizer (TV) of the SP-AMS was removed for this
study so that non-rBC-containing particles, which are not vaporized by the LV, were not detected. In contrast, the ToF-ACSM
only has a TV for aerosol vaporization and thus detects total NR-$PM_1$ components (i.e., Org, $NO_3^-$, $SO_4^{2-}$, $Cl^-$, $NH_4^+$) regardless
of their aerosol mixing state with rBC. The comparison of SP-AMS and ToF-ACSM measurements can provide quantitative
information on the degree of mixing between rBC and NR-$PM_1$ (Lee et al., 2017; Wang et al., 2017; Xu et al., 2018; Yu et al.,
2020).

The working principle of LV and the detailed description of SP-AMS has been reported by Onasch et al. (2012). In brief,
particles were focused by aerodynamic lens with an effective detection range between 70 and 1000 nm in diameter. rBC-
containing particles are heated up to ~4000 K and vaporized by the 1064 nm continuous Nd-YAG intra-cavity infrared (IR)
laser module. At such high temperatures, refractory coating materials (such as metals and certain inorganic salts) can also be
vaporized. The vapors are subsequently ionized by the 70 eV electron impact and then detected by a high resolution time-of-
flight mass spectrometer operated in V-mode with a mass spectral resolving power of ~2000 at m/z 28 (Decarlo et al., 2006;
Canagaratna et al., 2007). The particle vacuum aerodynamic diameter ($d_{va}$) was measured by the efficient particle time-of-
flight (ePToF) system. The SP-AMS was sampling at a 1-min interval for measuring bulk aerosol mass spectra and size
distributions, followed by a 2-min interval for event trigger single particle (ETSP) measurement. The ETSP data is not reported
in this work.

## 2.3 SP-AMS calibration

Regal Black (Regal 400R pigment black, Cabot Corp.) was used to calibrate the mass-based ionization efficiency of rBC
(mIE$_{rBC}$) for SP-AMS without the TV. Regal black particles were first suspended in ultra-pure water (Milli-Q, 18.2 MΩ. cm
(25°C) and total organic carbon (TOC) < 5 ppb). The suspension was sonicated continuously during atomization by a constant
output atomizer (model 3076, TSI Inc.). The droplets were then dried by a diffusion dryer and the dried particles were size
selected at 300 nm using a differential mobility analyzer (DMA, model 3081, TSI Inc.) for the SP-AMS calibration. The sum
of carbon ion clusters from $C_1^+$ to $C_{10}^+$ were used to quantify rBC mass for both ambient and calibration data. Due to the





significant interferences from OA, the average $C_1^+$-to-$C_3^+$ ratio of 0.66 obtained from Regal Black calibration was used to determine $C_1^+$ signals from rBC in ambient aerosol. Three sets of calibrations were performed throughout the sampling period.

Monodispersed and dried ammonium nitrate ($NH_4NO_3$) particles at 300 nm were generated using the atomization method
described above for determining the mass-based ionization efficiency of $NO_3^-$ ($mIE_{NO3}$). However, since pure $NH_4NO_3$ particles cannot be vaporized by the LV scheme, direct determination of $mIE_{NO3}$ was not possible for SP-AMS without TV. Therefore, $mIE_{NO3}$ and $mIE_{rBC}$ were first determined for our SP-AMS using the dual vaporization (DV, i.e., both TV and LV were installed) configuration. Noting that $mIE_{NO3}$ and $mIE_{rBC}$ were determined when the LV was off and on, respectively, with the TV operated at 600 ºC throughout the calibration. The relative ionization efficiency of rBC ($RIE_{rBC} = mIE_{rBC}/ mIE_{NO3}$) of
0.24 was then obtained. Assuming the $RIE_{rBC}$ does not change with the TV configuration, $mIE_{NO3}$ for the SP-AMS without TV can be calculated accordingly (i.e., $mIE_{NO3} = mIE_{rBC}/0.24$). The calculated $mIE_{NO3}$ and the default RIE values of $NO_3^-$ (1.1), $SO_4^{2-}$ (1.2), $Cl^-$ (1.3), Org (1.4), and $NH_4^+$ (4.0) (Jimenez, 2003) were used to quantify those NR-PM$_1$ coated on rBC core (referred to as NR-PM$_{coating}$ hereafter). We note that previous studies have suggested that this calibration approach may overestimate the mass of NR-PM$_{coating}$ (Willis et al., 2014; Lee et al., 2015).

Trace metals coated on rBC core can be detected by the LV configuration of SP-AMS (Carbone et al., 2015; Corbin et al., 2018; Rivellini et al., 2020). In this work, five metal-related ions, sodium ($Na^+$), potassium ($K^+$), Rubidium ($Rb^+$), Vanadium ($V^+$), and Nickel ($Ni^+$), were detected by the SP-AMS with their signals higher than the limit of detection during most of the sampling period (see Table S1). In addition, $K_3SO_4^+$ ions (m/z 213) originating from potassium sulfate were also observed and
will be discussed with the five metal-related ions. The signals of these six ions were not calibrated, and hence they are not included in the mass concentrations of NR-PM$_{coating}$ reported in this study. The raw signals (Hz) of these metal ions were used for most of the data analysis except for the source apportionment of OA and rBC that is based on positive matrix factorization (PMF) analysis of fragment ions detected by the SP-AMS (see Section 2.5 for detail).

**2.4 SP-AMS data processing**

The SP-AMS measurement was processed by the AMS data analysis software SQUIRREL, version 1.65C for unit mass resolution data, and PIKA, version 1.35C for high-resolution peak fitting (Sueper, 2015) . The daily particle-free ambient air was collected for determining the instrument lower detection limit and for correcting the fragmentation table to reduce interferences due to the air (Allan et al., 2004; Decarlo et al., 2006). Elemental ratios of OA coatings were determined using
the improved-ambient method (Canagaratna et al., 2015) with the conversion factor applied for oxygen-to-carbon (O:C) and hydrogen-to-carbon (H:C) ratios for LV measurements (Ma et al., 2021). The carbon oxidation state of OA ($OS_c$) was calculated based on the O:C and H:C ratios (i.e., $OS_c \approx 2*O:C – H:C$) (Kroll et al., 2011). The collection efficiency (CE) for rBC particles is largely governed by the overlapping between the particle beam and laser beam (Willis et al., 2014), depending



on particle shape. Willis et al. (2014) reported a CE value of 0.6 for bare BC but CE can increase with coating mass (i.e., NR-PM coating mass / rBC mass ratio, $R_{BC}$, greater than 1). The average $R_{BC}$ value for this study is $0.44 \pm 0.24$, and hence a constant CE value of 0.6 was applied in this work.

## 2.5 PMF analysis of SP-AMS data

PMF analysis has been widely used to identify various sources of ambient OA measured by aerosol mass spectrometers (Ulbrich et al., 2009; Zhang et al., 2011). In this work, the PMF evaluation tool (PET, v3.05D Ulbrich et al. (2009)) was used to identify potential sources and characteristics of rBC particles and their associated coatings. The OA and rBC ($C_n^+$) fragments were included for performing PMF analysis to form the base case in this work (PMF$_{base-n}$, where n represents the number of factors in the PMF solution selected for discussion). Similar to typical PMF analysis of OA fragments alone, ions with signal-to-noise (SNR) ratio between 0.2 and 2.0 were downweighed by a factor of 2. $CO_2^+$ related ions (i.e., $O^+$, $HO^+$, $H_2O^+$, and $CO^+$) and $C_1^+$ were also downweighed. Ions with SNR < 0.2 were removed from the analysis. This version of PMF analysis has been used in previous studies to better resolve the combustion sources at different locations (Massoli et al., 2015; Lee et al., 2017; Wang et al., 2017; Carbone et al., 2019; Rivellini et al., 2020).

Given that metal ions are relatively inert in response to chemical aging of atmospheric aerosol, integrating metal ions in PMF may better resolve the origin and emission characteristics of rBC-containing particles in complex urban environments. The metal-related ions (i.e., $Na^+$, $K^+$, $Rb^+$, $V^+$, $Ni^+$, and $K_3SO_4^+$) were included in our PMF analysis (denoted as PMF$_{metal-n}$ hereafter). Due to the large uncertainties of RIE for metals, the PMF$_{metal}$ results would only be used for identifying potential origins of OA and the metal concentrations were not included in the total mass concentration of each PMF factor. Signal of $K^+$ was downweighed by 10 times, whereas $Rb^+$, $V^+$, and $Ni^+$ were upweighted by a factor of 3 the PMF solution. Previous studies of SP-AMS measurements have also used a similar approach to integrate metal-related ions into PMF analysis (Carbone et al., 2019; Rivellini et al., 2020). To further investigate the nature and formation of OA, IA-related fragments from sulfate, nitrate, and chloride were included in the PMF analysis. This PMF analysis included all the measured aerosol components (i.e., OA + rBC + metal + IA) (denoted as PMF$_{all-n}$ hereafter). Sulfate fragments were downweighed by a factor of 3 due to their relative strong intensities.

The RIE value of 1 was used for all the species applied in the PMF analysis and their mass concentrations were determined using their RIE values for each factor described in Section 2.3. The PMF results were evaluated for up to 10 factors with F peak values between -1 to 1 with a step change of 0.2. As noted below, the first three days of the sampling period had elevated concentrations of rBC-containing particles due to significant influences of regional transport. Removing this short period does not make significant changes in the PMF results. The Q/Q$_{expected}$ plots for different PMF scenarios up to 10-factor solution are shown in Fig. S1. The detailed interpretation of selected solution for each PMF input (i.e., PMF$_{base-n}$, PMF$_{metal-n}$, and PMF$_{all-n}$)



will be discussed in detail in Section 3.2-3.5. Factors from each solution will be denoted as "Factor name"$_{solution-n}$ (e.g., Factor-A identified in PMF$_{base-4}$ will be referred as Factor-A$_{base-4}$).

## 2.6 Wind and air mass back trajectory analysis

The potential sources of pollutants were estimated using ZeFir nonparametric wind regression (NWR) described by Petit et al. (2017) using Igor Pro (Ver. 8.04, WaveMetrics Inc.). To achieve potential sources of pollutants, both high resolution temporal wind information (i.e., wind speed and direction) and pollutant concentration are required to calculate nearby sources (Henry et al., 2009). In this study, the NWR graphs were generated for an angle resolution of 1° and radial resolution of 0.1 m s$^{-1}$ with program-suggested smooth parameters of 12 and 2.21, respectively. The origins of air masses were generated by the Hybrid

Single-Particle Lagrangian Integrated Trajectory (HYSPLIT) model coupled with meteorological data from the Global Data Assimilation System (GDAS, 1∘) for 5 days back for every hour. This model was developed by the National Oceanic and Atmospheric Administration (Draxler and Hess, 1998). Cluster analysis was also performed using the HYSPLIT model with details described by previous studies (Borge et al., 2007; Baker, 2010). Potential source contribution function (PSCF) was used to analyze regional transport and sources to expand the source knowledge to a larger scale (Polissar et al., 2001). The

PCSF model re-distributes pollutant concentration or hotspots based on the air mass trajectories' residence times. In this work, PCSF model were set to consider the species concentration above their 75$^{th}$ percentile with a cell size of 0.1° and smoothing of 2.

## 3.   Results and discussion

### 3.1 Measurements overview

Figure 2a-c presents the meteorological parameters (solar radiation, precipitation, RH, temperature, wind speed, and wind direction) during the measurement period. The average temperature and RH were 28 °C and 85%, respectively, and reached their daily maximum (~31°C) and minimum (~70% RH) at ~13:00 LT (Fig. 3a) The sampling period covered the Northeast monsoon season, which is supported by the cluster analysis of 72-hours air mass back trajectory that showed regional transport

of air mass from Northeast of Singapore dominated the campaign period (Fig. S2) where multiple fires were observed (Fig. S3). No precipitation was observed for most of the sampling period with only a few days having rainfall higher than 1 mm. In general, solar radiation was strong during daytime that could lead to oxidized secondary coating formation on rBC particles (i.e., Org/rBC, R$_{BC}$, and O:C of organic coating peaked at ~13:00 LT) as illustrated in Fig. 3b and c. The wind was blown from the north-west for the majority of sampling period and generally shifted to southwest/southeast in the afternoon which could

carry shipping and industrial emissions from the coastal area to the measurement site (Figure 3d). Detailed NWR and PSCF analysis for individual species and PMF factors will be provided in Sections 3.2-3.5 for identifying local emissions and regional transport of aerosol components.





Figure 2. Time series of (a) solar radiation (SR) and precipitation, (b) RH and temperature, (c) wind direction and wind speed, (d) $NO_x$, $O_3$ and rBC concentrations, (e) Org, $SO_4^{2-}$, $NO_3^-$, $Cl^-$ and $NH_4^+$ concentrations, and (f) metal-related ions intensities.

5 The pie charts in panels e and f show the overall mass distribution of rBC and NR-PM$_{coating}$ in rBC-containing particles and metal-related ions signal distribution (Hz) in rBC-containing particles, respectively.





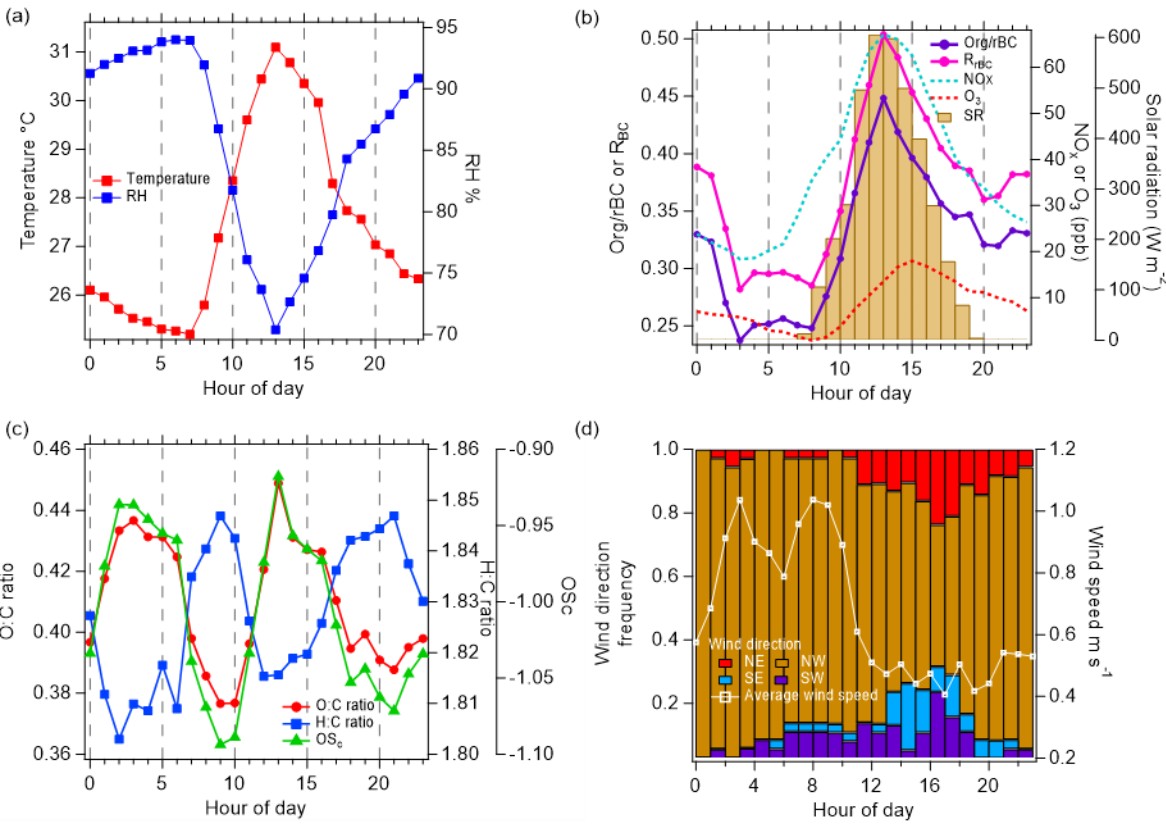

Figure 3. Diurnal cycles of (a) temperature and RH, (b) Org/rBC ratio, $R_{BC}$, $NO_x$, $O_3$, and solar radiation (SR), (c) O:C, H:C and $OS_c$ of organic coating on rBC particles, and (d) wind direction fraction distribution and wind speed.

The time series of chemical components of rBC-containing particles measured by the SP-AMS are shown in Figure 2d-f. The average total mass concentration of rBC plus NR-PM$_{coating}$ was $6.37 \pm 4.15$ µg m$^{-3}$. rBC was the major component ($4.47 \pm 2.89$ µg m$^{-3}$), accounting for ~70 wt% of rBC-containing particles. The main contributor of NR-PM$_{coating}$ was Org ($1.67 \pm 1.44$ µg m$^{-3}$) that contributed to ~26 wt% of the rBC-containing particles. The IA components ($SO_4^{2-} = 0.16 \pm 0.16$ µg m$^{-3}$, $NO_3^- = 0.06 \pm 0.05$ µg m$^{-3}$, $Cl^- = 0.02 \pm 0.01$ µg m$^{-3}$, and $NH_4^+ = 0.02 \pm 0.01$ µg m$^{-3}$) were minor contributors to the NR-PM$_{coating}$ mass (Fig.

2e). The rBC concentrations measured by the SP-AMS was about 31% higher than the BC concentrations measured by the aethalometer, but a strong temporal correlation between the two measurements was observed (R = 0.89, Fig. S4). Such strong correlation between the two instruments was consistently observed at the same sampling location in previous work (Rivellini et al., 2020). The time series correlation of SP-AMS and ToF-ACSM for Org, $SO_4^{2-}$, $NO_3^-$, $Cl^-$, and $NH_4^+$ are shown in Fig. S4b-f. Strong correlations were observed for all species (R > 0.74) except for ammonium (R = 0.31). Based on the SP-AMS

and ToF-ACSM measurements (i.e., NR-PM$_{coating}$ vs. NR-PM$_1$), ~15% of OA and less than 5% of each IA component were coated on rBC particles (Fig. S4), suggesting that most of the NR-PM$_1$ materials were externally mixed with rBC in this study.





The correlation coefficients between all the species measured by the SP-AMS are reported in Fig. S5. As the major component of NR-PM$_{coating}$, Org had strong correlations with rBC (R = 0.8), implying that large fractions of coated Org are co-emitted primarily with rBC from various combustion sources. In contrast, SO$_4^{2-}$, NO$_3^-$, Cl$^-$, and NH$_4^+$ are only weakly or moderately

correlated with rBC (R = 0.32-0.53), which is consistent with the fact that these IA components are generally secondarily formed in the atmosphere. Previous measurement in Singapore showed that SO$_4^{2-}$ can be the second largest NR-PM$_1$ contributor (~38% NR-PM$_1$ mass) (Rivellini et al., 2020), and with our observation that rBC particles were not the major condensation sink for the secondary IA components (i.e., less than 5%, Fig. S4). Regarding the metal-related ions, Na$^+$, K$^+$ and K$_3$SO$_4^+$ gave much stronger signals compared to V$^+$, Ni$^+$ and Rb$^+$ (Fig. 2f). Among all the detected metal-related ions, Na$^+$

had the strongest correlations with rBC (R = 0.73), followed by V$^+$ (R = 0.58) and Ni$^+$ (R = 0.44), whereas K$^+$, Rb$^+$, and K$_3$SO$_4^+$, were only weakly correlated with rBC (R < 0.3). The wide range of correlation coefficients between rBC and different metals suggests multiple combustion sources of rBC particles with different metal contents observed at the measurement site (see Section 3.2.1).

## 3.2 Characteristics of rBC and associated coating materials

### 3.2.1 Characteristics of metal ions associated with rBC-containing particles

Figure 4a shows that the mass loadings of rBC remained low at night and increased during the morning and evening rush hours (8:00-9:00 and 20:00 LT), indicating that local traffic is one of the key contributors to rBC. Furthermore, rBC concentrations started increasing at ~12:00 LT and peaked at 17:00-18:00 LT along with Org, SO$_4^{2-}$, NO$_3^-$, Na$^+$, Ni$^{+,}$ and V$^+$. The afternoon

peak of these species could be partially due to the wind from south direction (Fig. 3d) that could carry emissions from shipping ports and/or the industrial region to the measurement site occasionally (Fig. 1) (Betha et al., 2013; Budisulistiorini et al., 2018; Jaafar et al., 2018; Rivellini et al., 2020). A distinct hotspot was observed at the south of our measurement site for Na$^+$, V$^+$ and Ni$^+$ under relatively high wind speed condition (> 8 m/s, Fig. 4g-i), and their concentrations were not significantly influenced by regional transport as shown in their PSCF plots (Fig. 5). Assuming a similar RIE for V$^+$ and Ni$^+$, the average V$^+$/Ni$^+$ ratios

at the afternoon peak was approximately equal to 4-5 (with 75[th] percentile > 6, Fig. 6a). These observations suggest that these metal ions can be used to indicate the impacts of shipping and/or industrial emissions on specific rBC and OA sources identified in the PMF analysis (see Section 3.3).



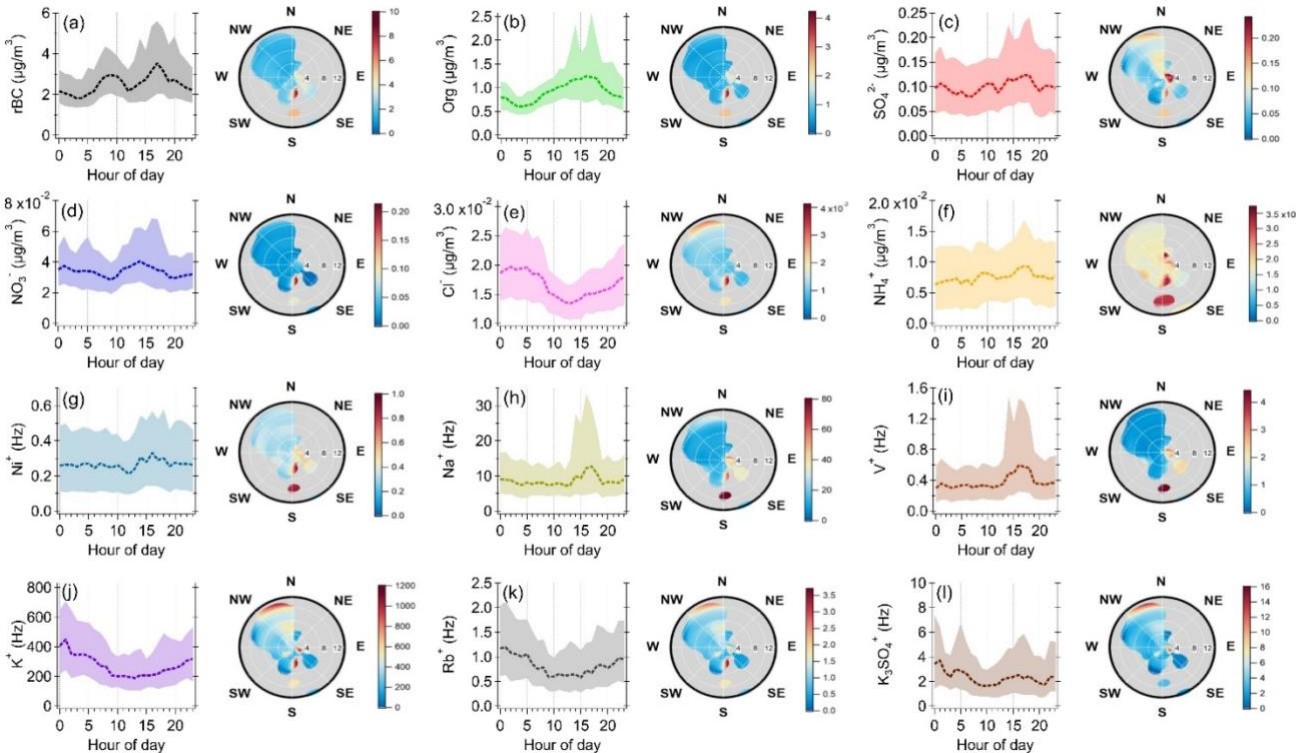

Figure 4. Diurnal cycles and NWR plots of individual species measured by SP-AMS: median (dashed line), $25^{th}$ and $75^{th}$ percentile (shaded areas). Mass concentrations of organic and inorganic species are corrected by CE and RIE except for metal ions and $K_3SO_4^+$ that are shown in signal intensity (Hz). The wind speed unit in NWR plots (radial axis) is m/s. The color scale
of NWR plots represents species concentration or signal intensity (μg/m³ or Hz).

A strong correlation was observed between $K^+$ and $Rb^+$ (R = 0.75, Fig. S5), and they had relatively high concentrations at midnight (Fig. 4j and k). Although potassium and rubidium are not unique tracers for a specific combustion source, our previous SP-AMS measurements with DV scheme at the same site has reported a strong temporal correlation between these
two ions, and has highlighted that $K^+$ and $Rb^+$ can be used to understand the impacts of regional biomass burning on aged background SOA (Rivellini et al., 2020). $K_3SO_4^+$ is strongly correlated with $K^+$ (R = 0.83, Fig. S5) in this work. Detectable signals of $K_3SO_4^+$ indicate the presence of potassium sulfate ($K_2SO_4$). Potassium salts generated from biomass burning are typically in the form of potassium chloride (KCl), which can react with acid gases (e.g., $H_2SO_4$ and $HNO_3$) heterogeneously to form $K_2SO_4$ and potassium nitrate ($KNO_3$) during the aging process (Li et al., 2003; Li et al., 2010). Previous SP-AMS
measurements have reported the detection of $K_3SO_4^+$ in rBC-containing particles in Beijing and Tibetan Plateau and have used $K_3SO_4^+$ as an indicator for aged or transported biomass burning organic aerosol (BBOA) (Wang et al., 2020). In addition to SP-AMS measurements, Murphy and Thomson (1997) was able to detect $K_3SO_4^+$ from plume contained biomass burning





pollution using positive ion laser ionization single particle analysis. $K_2SO_4$ has also been detected as potassium ion clusters, including $K_3SO_4^+$ and $K_2HSO_4^+$, in aged biomass burning particles by the positive mode of aerosol time-of-flight mass spectrometers (ATOFMS) (Zauscher et al., 2013).

The PSCF analysis and NWR plots show that the elevated signals for $K^+$, $Rb^+$ and $K_3SO_4^+$ are associated with the regional transport of air masses from north of Singapore (Fig. 4 and 5), indicating the potential large influence of regional biomass burning emissions with different degree of aging on the chemical composition of NR-PM$_{coating}$, and the high intensities were mainly observed at the beginning of our study period (Jan 29-31, 2022, Fig. 2f). Figure S3 shows the regional fire events observed during Jan 22-29, 2022. Furthermore, the diurnal cycle of $K_3SO_4^+$/ $K^+$ ratio illustrates additional sources of $K_3SO_4^+$

in the afternoon compared to $K^+$ (i.e., the ratio peaked at ~16:00 LT, Fig. 6b) during which the NR-PM$_{coating}$ was most acidic on average (i.e., lower measured-to-predicted $NH_4^+$ ratio ($NH_4^+_{meas}$/$NH_4^+_{pred}$) at ~16:00-17:00 LT, Fig. 6c) that might facilitate heterogeneous conversion of background potassium salts to $K_2SO_4$ in a relatively short time scale (see Section 3.2.3 for acidic sulfate formation). Overall, the above observations suggest that $K^+$, $Rb^+$, and $K_3SO_4^+$ can be potentially useful to indicate the relative importance of fresh and aged biomass burning emissions on specific rBC and OA sources identified in the PMF

analysis (see Section 3.3).

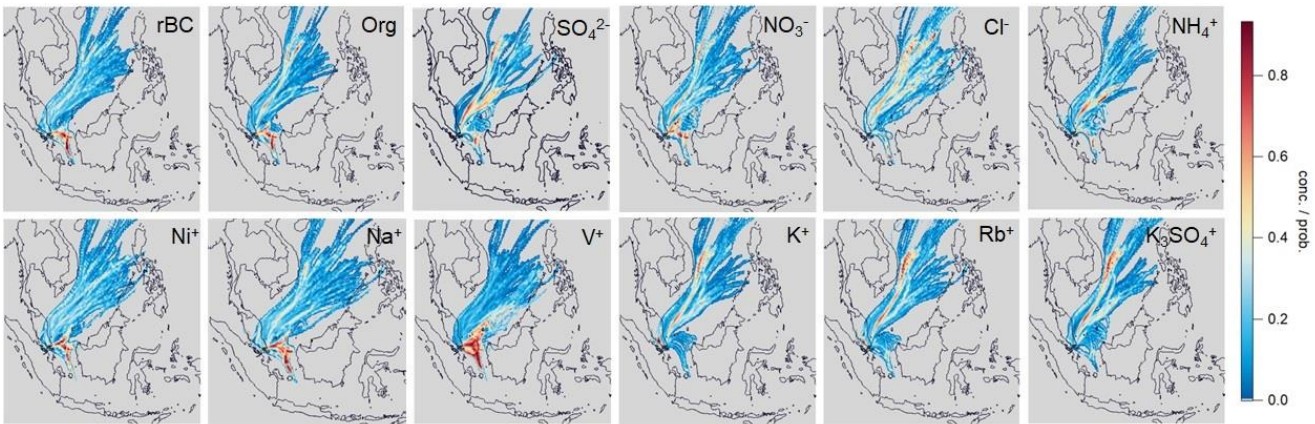

Figure 5. PSCF plots for individual species measured by SP-AMS. PCSF model were set to consider the species concentration above their 75th percentile.



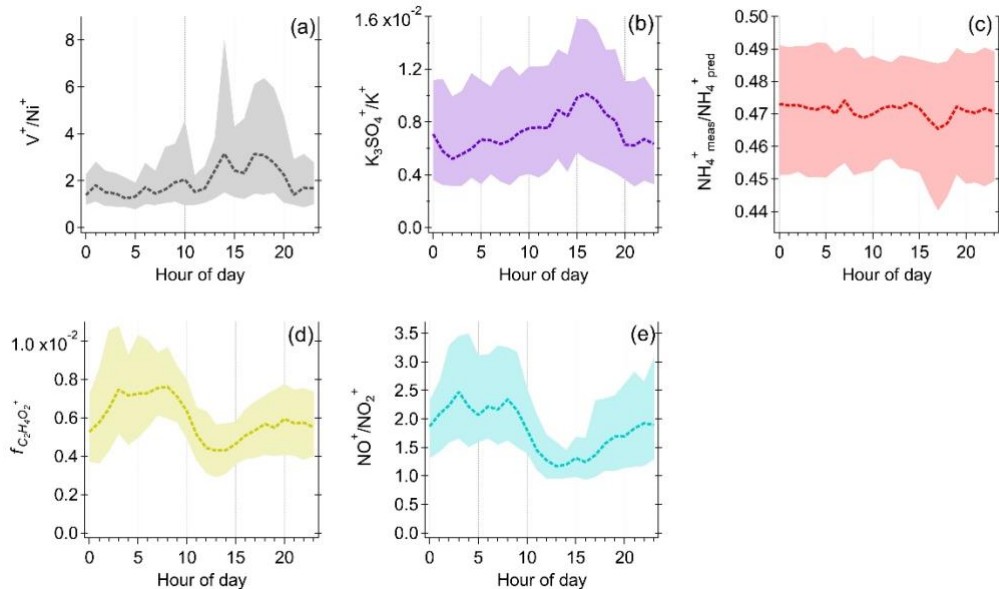

Figure 6. Diurnal cycles of (a) $V^+/Ni^+$, (b) $K_3SO_4^+/K^+$, (c) $NH_4^+{}_{meas}$ / $NH_4^+{}_{pred}$, (d) $f_{C_2H_4O_2^+}$ and (e) $NO^+/NO_2^+$ measured by SP-AMS.

### 3.2.2 Characteristics of OA components in NR-PM$_{coating}$

OA was the major component of NR-PM$_{coating}$, accounting for ~26 wt% of the rBC-containing particles. The organic-to-rBC (Org/rBC) ratio and elemental ratios (i.e., O:C, H:C and OS$_c$) are useful parameters for identifying the nature of OA coating (e.g., POA vs. SOA). As shown in Fig. 3b and c, the diurnal patterns of Org/rBC, O:C and OS$_c$ have lower values, indicative of POA, at 8:00-9:00 and 20:00-21:00 LT, during which the local traffic emissions are the primary sources of rBC (see Section 3.2.1). This observation indicates that the rBC emitted from local traffic was mainly coated by POA and the thickness of such POA coatings was relatively thin for the majority of the rBC particles compared to those observed during the rest of the day. The Org/rBC, O:C and OS$_c$ started increasing significantly at 10:00 LT and their peaks were observed at 12:00-13:00 LT when the solar radiation was the strongest. This observation suggests that condensation of photochemically produced SOA can lead to notable increase of OA coating thickness, although most of the OA in NR-PM$_1$ was externally mixed with rBC particles (see Section 3.1).

The Org/rBC ratio declined in the afternoon and then remained roughly the same between 18:00 LT and 2:00 LT next day. The relatively high O:C ratio and OS$_c$ overnight suggests that the OA coatings were largely contributed by oxidized organic species, which is consistent with greater influence from regional transport of aged aerosols with biomass burning influences





overnight, as discussed above in Section 3.2.1. To investigate the possible biomass burning influences on OA coatings, a mass contribution of a biomass burning tracer fragment to the total organic coating ($f_{C_2H_4O_2^+}$) was determined as shown in Fig. 6d. Cubison et al. (2011) reported that $f_{C_2H_4O_2^+} > 0.3\%$ obtained from standard AMS measurements can be the indication of biomass burning influences. Ma et al. (2021) showed that $f_{C_2H_4O_2^+}$ can be enhanced by a factor of ~2.45 on average for

oxygenated organic coating and 2.33 for levoglucosan vaporized by the LV scheme of SP-AMS. The higher values of $f_{C_2H_4O_2^+}$ and $OS_c$ of OA coating materials between 2:00 and 8:00 LT still provide additional evidence of stronger influences of aged/oxidized biomass burning emissions on the chemical characteristics of rBC-containing particles during the night.

### 3.2.3 Characteristics of IA components in NR-PM$_{coating}$

Although NO$_3^-$, SO$_4^{2-}$, Cl$^-$, and NH$_4^+$ together accounted for less than 4 wt% of the rBC-containing particles (Fig. 2e), their temporal variations can still be useful for evaluating the potential sources of rBC-containing particles and the atmospheric processes that the particles might have experienced. The elevated concentrations of SO$_4^{2-}$ and NO$_3^-$ together with rBC and a few metal ions between 14:00 and 19:00 LT (Fig. 4) suggest potential impacts of atmospheric processing of shipping and/or industry emissions on SO$_4^{2-}$ and NO$_3^-$ concentrations (Section 3.2.1). The average NH$_4^+$$_{meas}$ /NH$_4^+$$_{pred}$ (Zhang et al., 2007) was

~0.47 most of the time and dropped to the lower values between 16:00 and 18:00 LT (Fig. 6c), implying that acidic aerosol were locally formed via photooxidation that subsequently condensed on rBC particles and they could not be completely neutralized by gas-phase ammonia before they reached the site. A similar observation has been reported for NR-PM$_1$ based on the SP-AMS measurement with DV scheme at the same site (Rivellini et al., 2020). The SO$_4^{2-}$ concentration is the highest followed by NO$_3^-$ and Cl$^-$ concentrations (38% and 13% of SO$_4^{2-}$ concentration, respectively) so SO$_4^{2-}$ is likely the major

component affecting the aerosol acidity. The oxidation of biogenic dimethyl sulfide (DMS) from ocean can produce methanesulfonic acid (MSA) that might contribute to the detected sulfate-related fragments at our sampling site (Rivellini et al., 2020) but the signature peaks of MSA and organosulfur compounds, including CH$_3$SO$^+$, CH$_2$SO$_2^+$, and CH$_3$SO$_2^+$ (Farmer et al., 2010; Ge et al., 2012; Saarikoski et al., 2019; Rivellini et al., 2020), were not detected in this work.

NO$_3^-$ and Cl$^-$ have elevated levels over midnight (i.e., peaked at around 1:00 LT) along with K$^+$, Rb$^+$ and K$_3$SO$_4^+$, suggesting that these IA components might condense on biomass burning-influenced rBC-containing particles at lower ambient temperature during nighttime. Nevertheless, it is important to point out that NO$^+$/NO$_2^+$ ratio (i.e., the two major fragments for quantifying NO$_3^-$) have been widely used to determine the presence of organo-nitrate based on the standard AMS measurement as the ratios of organo-nitrate are generally higher than that of inorganic nitrate (Farmer et al., 2010; Xu et al., 2015; Ng et al.,

2017). Configuring the SP-AMS with LV scheme alone, the average NO$^+$/NO$_2^+$ ratio for NH$_4$NO$_3$ coated on regal black particles generated in the laboratory is ~0.9. The higher NO$^+$/NO$_2^+$ ratios (~2-3.6) between 19:00 and 11:00 LT (Fig. 6e) suggests the existence of organo-nitrate in this work. On the contrary, the lower NO$^+$/NO$_2^+$ ratio during daytime suggests




photochemical production of inorganic nitrate was more important than that of organo-nitrate. The large similarity between the diurnal patterns of $f_{C_2H_4O_2^+}$ (Fig. 6d) and $NO^+/NO_2^+$ ratio (Fig. 6e) indicates that biomass burning aerosol can be a possible contributor of organo-nitrate during the sampling period (Lin et al., 2016; Lin et al., 2017). Organo-nitrate can be produced via nocturnal $NO_3$ radical and $N_2O_5$ chemistry (Fry et al., 2018). A laboratory study has shown that under higher RH condition

(50% vs. 10%), $NO_3$ radicals formed from the dark reaction of $O_3$ and $NO_2$ can promote up to 70% more SOA mass formation with BBOA (Kodros et al., 2020), suggesting that aged BBOA can be a significant secondary source for organo-nitrate through nighttime chemistry.

### 3.2.4 Size distribution of rBC and associated coating materials

Figure 7a presents the size distributions of rBC and the major species of NR-PM$_{coating}$ (i.e., OA, $NO_3^-$, and $SO_4^{2-}$). rBC mass peaked at around 180 nm, and its concentration gradually decreased as a function of particle size. This observation implies that the majority of rBC mass was emitted by local combination sources without substantial degree of atmospheric aging, which is well aligned with the results of source apportionment analysis (see below) that ~81% of rBC was emitted from local traffic and industrial emissions. Bimodal distribution was clearly observed for OA, and the intensities of peaks at ~200 nm and

465 nm are comparable. This implies that OA in NR-PM$_{coating}$ were influenced significantly by both primary and secondary sources, which is consistent with the results of source apportionment analysis that ~41% and 49% of OA in NR-PM$_{coating}$ was due to local primary emissions and SOA, respectively. $NO_3^-$, and $SO_4^{2-}$ peaked between 500 and 600 nm, suggesting that most of these IA species were associated with more aged rBC-containing particles. Nevertheless, a weaker peak of $NO_3^-$, and $SO_4^{2-}$ at ~200 nm highlights that these IA might condense on freshly emitted rBC in a relatively short time scale when they were

locally formed via photochemistry. The details of source apportionment results based on different PMF scenarios will be discussed in Section 3.3.

The size distributions of some fragment ions, including m/z 43, 44, 55, 57, and 60, are shown in Fig. 7b, and most of them also had bimodal distributions. Signals of m/z 43, 55, and 57 could be significantly contributed by both $C_xH_y^+$ and $C_xH_yO_z^+$

fragments from OA, but their relative contributions across different particle size cannot be resolved by the unit mass resolution (UMR) data. It is reasonable to argue that the smaller size OA (at ~200 nm) were dominated by $C_xH_y^+$ fragments as they mainly co-emitted with rBC from local combustions as demonstrated by the mass spectral profiles of traffic emissions reported in Section 3.3. In contrast, $C_xH_yO_z^+$ fragments were likely the major contributor to the larger size OA (at ~465 nm) due to atmospheric aging and SOA formation. $C_2H_3O^+$ at m/z 43 has been shown as the major fragments for laboratory-generated and

ambient oxygenated OA materials (Lee et al., 2017; Ma et al., 2021). This argument is further supported by our observation that a stronger peak was observed at larger size particles for m/z 44, which is mainly due to $CO_2^+$ fragments from ambient oxygenated OA. We note that m/z 44 could be also influenced by the refractory oxygenated functionality of soot surface structure (Corbin et al., 2015; Ma et al., 2023), especially for the rBC-containing particles with lower Org/rBC ratio observed



at the smaller size (Fig. 7b) (Lee et al., 2015; Willis et al., 2016; Lee et al., 2017). Although m/z 60 can be mainly contributed by $C_5^+$ from rBC and $C_2H_4O_2^+$ from OA (i.e., the biomass burning tracer organic fragment), the size distribution of rBC, m/z 39 ($K^+$) and m/z 213 ($K_3SO_4^+$) (Fig. 7b and c) suggests that biomass burning-influenced rBC-containing particles observed in this work had relatively large particle size and $C_2H_4O_2^+$ is likely the key contributor to the m/z 60 signals within the larger size range.

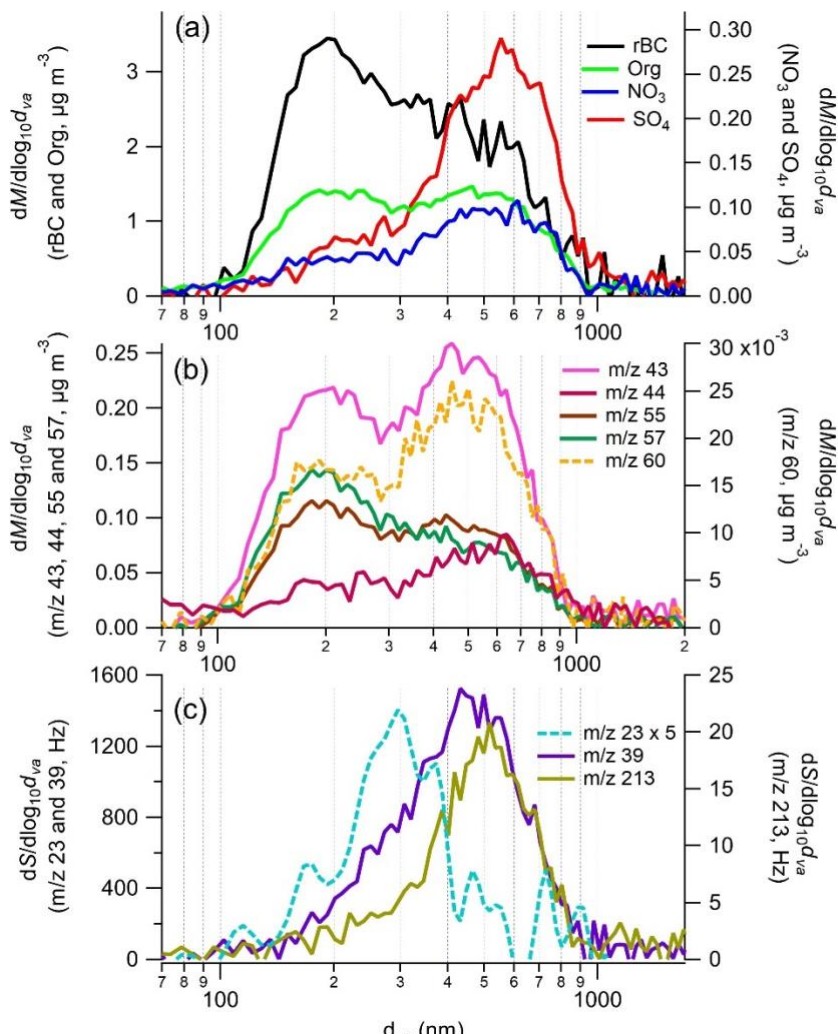

Figure 7: Size distribution of rBC and associated coating materials. Signals at m/z 43, 44, 55, 55, and 60 were converted to mass concentrations of organics assuming they are mainly contributed by OA. Signals at m/z 23, 39 and 213 are used to represent $Na^+$, $K^+$ and $K_3SO_4^+$, respectively. Signals at m/z 23 were smoothed and multiplied by a factor of 5.





### 3.3 Source apportionment of rBC and OA coatings based on PMF$_{base}$

In this section, source apportionment analysis of rBC and OA coatings is presented, and the PMF$_{base-4}$ results are considered
as a base case for comparison in the later sections. Figure 8 shows mass spectra, diurnal cycles and time series of the PMF$_{base-4}$ solution determined by OA and rBC fragments. The major types of particles identified in the PMF$_{base-4}$ were rBC-rich, hydrocarbon-like OA (HOA), less-oxidized oxygenated OA (LO-OOA), and more-oxidized oxygenated OA (MO-OOA) factors. The rBC-rich$_{base-4}$ factor is the major rBC contributor (~80%) while OA mass was more evenly distributed among the four factors, with the rBC-rich$_{base-4}$, HOA$_{base-4}$, LO-OOA$_{base-4}$, and MO-OOA$_{base-4}$ factors accounted for 17%, 31%, 21%, and
31% of total OA, respectively (Fig. 9). Cooking-related OA (COA) was not observed in this work, but was clearly identified using SP-AMS operated at DV mode at the same location due to the kitchen emissions from the canteens and residential area nearby (Rivellini et al., 2020). This comparison suggests that COA was externally mixed with rBC at this site, which has been previously observed in other urban environments (Willis et al., 2016; Lee et al., 2017; Farley et al., 2024).

### 3.3.1    POA from fossil fuel combustion

The HOA$_{base-4}$ factor was dominated by aliphatic fragments ($C_xH_{2x-1}^+$ and $C_xH_{2x+1}^+$) with the highest mass contribution to OA in NR-PM$_{coating}$ (31%). The rBC fragments accounted for only ~13.2% of HOA$_{base-4}$ mass, resulting in Org/rBC ratio of 6.6. The low O:C ratio of this factor (0.14) suggests the primary nature of HOA coating. The rBC-rich$_{base-4}$ factor was dominated by $C_1^+$ - $C_5^+$ fragments with the highest mass contribution to total rBC (80%). The low Org/rBC ratio of rBC-rich$_{base-4}$ factor
(0.37) suggests that most of the rBC-containing particles are thinly coated with OA at our measurement site. The rBC-rich$_{base-4}$ factor has a $C_1^+/C_3^+$ ratio of 0.67 that is similar to that of Regal Black standard and in line with the range of rBC emitted from diesel engine or locomotive engine emissions (Onasch et al., 2012; Corbin et al., 2014; Carbone et al., 2019). The $CO^+$ and $CO_2^+$ fragments accounted for 38% of OA mass in the rBC-rich$_{base-4}$ factor. These two oxygenated fragments can be due to the presence of oxidized organic compounds and soot functionality, and hence the mass and O:C ratio of OA coating may be
overestimated in the rBC-rich$_{base-4}$ factor. $C_3O_2^+$ is another oxygenated rBC fragment that has been observed in refractory soot structures of laboratory-generated diesel soot particles (Ma et al., 2023). The $C_3O_2^+/C_3^+$ signal (Hz) ratio observed in the rBC-rich$_{base-4}$ factor is 0.094. Assuming all $CO^+$, $CO_2^+$ and $C_3O_2^+$ signals from rBC, the Org/rBC ratio drops to ~0.1 for rBC-rich$_{base-4}$.

The HOA$_{base-4}$ and rBC-rich$_{base-4}$ factors had similar diurnal profiles (Fig. 8), and strongly correlated with bulk rBC concentrations (R = 0.99 and 0.92, respectively), indicating that they represent rBC and POA components emitted from local combustion sources. The NWR plots also show that the HOA$_{base-4}$ and rBC-rich$_{base-4}$ factors shared similar emission hotspots when the wind speed was greater than 8 m/s from the south direction. Similar to the diurnal pattern of rBC, both factors have



typical rush hour peaks at ~09:00 and 20:00 LT that were previously reported by Rivellini et al. (2020) at the same measurement site. Outside of the rush hours, both factors started increasing at ~12:00 LT and peaked at 17:00-18:00 LT that coincided with prevailing wind blown from the industrial and shipping region. Although most of the previous studies assigned their HOA and rBC-rich factors as local traffic-related OA (Gentner et al., 2017; Malmborg et al., 2017; Carbone et al., 2019), our results suggest that rBC-rich$_{base-4}$ and HOA$_{base-4}$ can be emitted from multiple combustion sources in this work. Nevertheless, the PMF$_{base}$ could not further resolve the contributions of traffic and shipping/industrial emissions to these two primary OA factors.

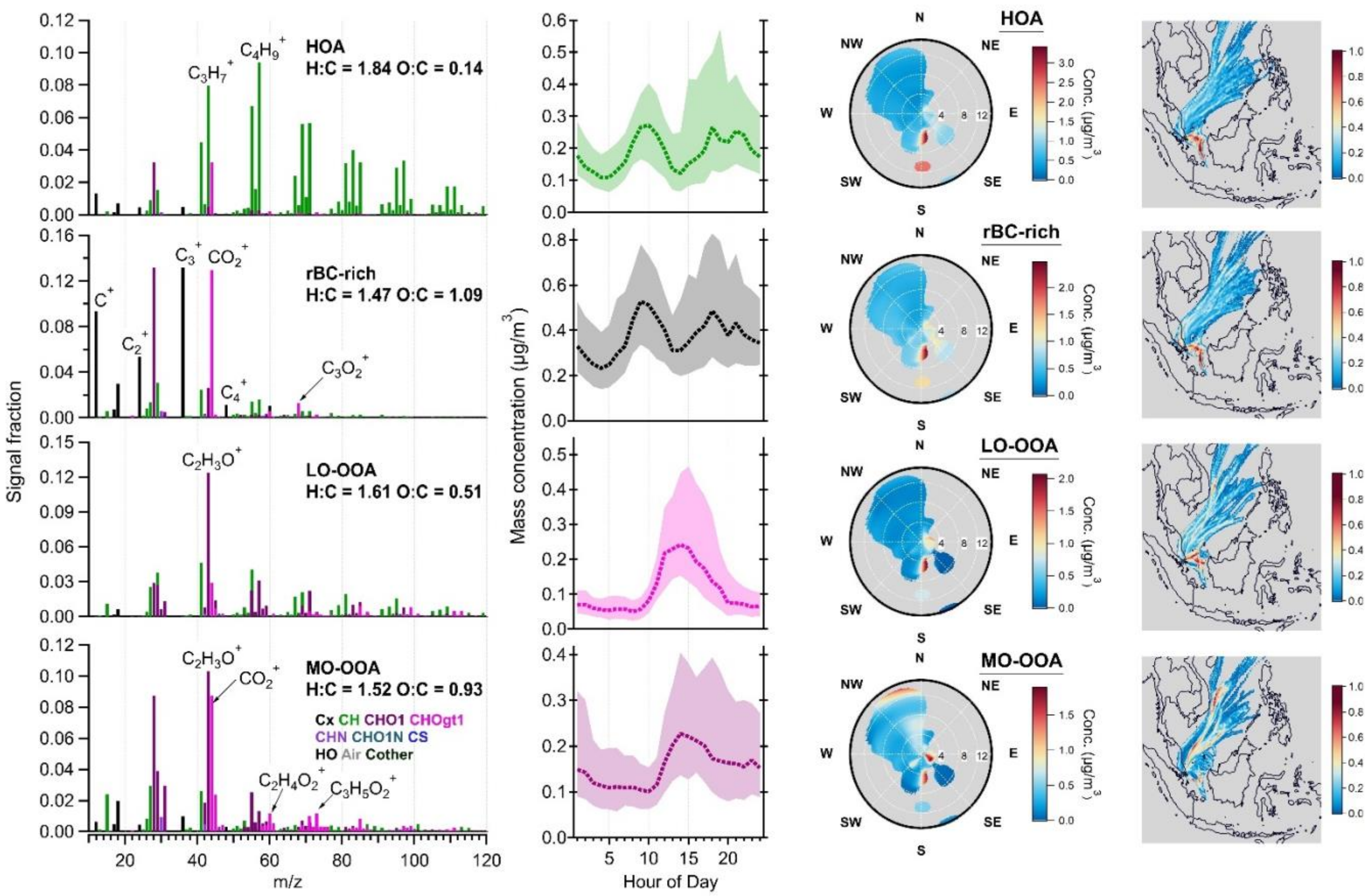

Figure 8. Results of PMF$_{base-4}$ factors: Mass spectra of OA and rBC (1st column), diurnal cycles of mass concentrations (2nd column, median: dashed lines, 25th/75th percentiles: shaded areas), NWR plots (3rd column), and PSCF plots (4th column). PMF mass spectrum were not RIE adjusted for different species. The wind speed unit in NWR plots (radial axis) is m/s. The color scale of NWR plots represents species concentration (µg/m$^3$). Some of the NWR plots presented with different wind speed ranges (< 3 m/s) and resolution are shown in Fig. S6.



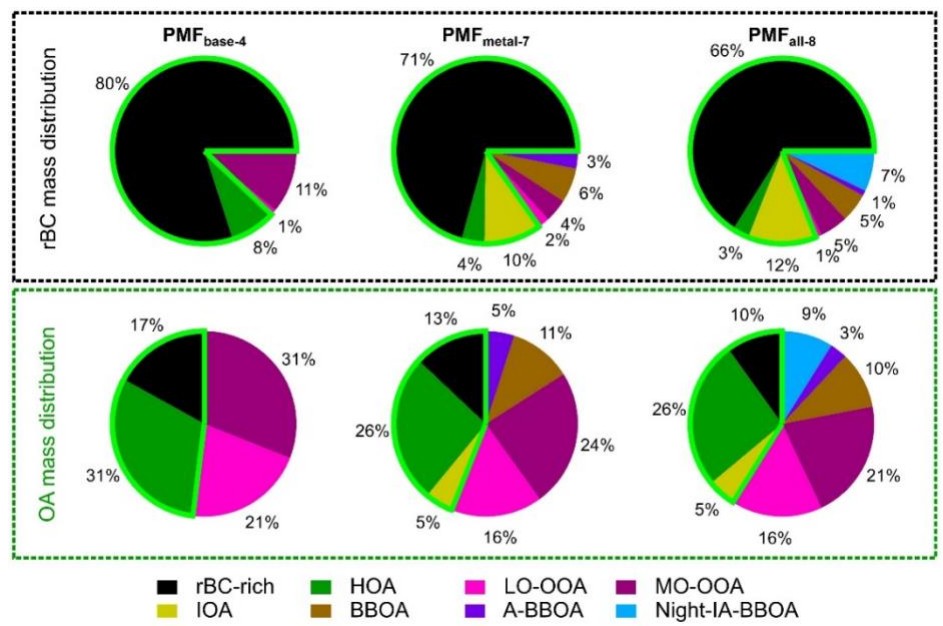

Figure 9. rBC (top panel) and Org (bottom panel) mass distributions for different PMF factors: PMF$_{base-4}$ (left column), PMF$_{metal-7}$ (middle column), and PMF$_{all-8}$ (right column). The light green boundary in each pie chart indicates the total contribution of PMF factors that are related to primary emissions from anthropogenic combustion.

### 3.3.2 SOA from local chemistry and regional transport

Two secondary OA factors, namely LO-OOA and MO-OOA, were identified by PMF$_{base-4}$. The mass spectra of LO-OOA$_{base-4}$ were dominated by a $C_2H_3O^+$ fragment with comparable contributions between other $C_xH_y^+$ and $C_xH_yO_z^+$ fragments. LO-OOA$_{base-4}$ had an O:C ratio of 0.51, which is more oxygenated than the HOA$_{base-4}$ factors. The diurnal profile of LO-OOA$_{base-4}$ increased during daytime together with solar radiation (Fig. 8 and 3b). The PSCF plot of LO-OOA$_{base-4}$ (Fig. 8) also suggests that the elevated concentrations of LO-OOA$_{base-4}$ were not strongly associated with regional transport of air mass. Given that rBC only accounted for 1.6% mass of LO-OOA$_{base-4}$, the high Org/rBC ratio of LO-OOA$_{base-4}$ (61.0) suggests that the LO-OOA$_{base-4}$ factor likely represents freshly formed SOA due to local photooxidation chemistry that are thickly coated on rBC particles within a relatively short time scale. Noting that this hypothesis remains valid even though the Org/rBC ratios for LO-OOA reduces to ~18 in PMF$_{metal}$ and PMF$_{all}$ scenarios (See Sections 3.4 and 3.5 and Table S2), which is still considered as thickly coated rBC-containing particles compared to other ambient observations (Cappa et al., 2019).

The mass spectra of MO-OOA$_{base-4}$ were dominated by $C_2H_3O^+$ and $CO_2^+$ fragments. The MO-OOA$_{base-4}$ factor has the O:C ratio of 0.93, and hence MO-OOA$_{base-4}$ likely represents the more oxygenated SOA components compared to LO-OOA$_{base-4}$.





rBC accounted for 12.8% of MO-OOA$_{base-4}$ mass with the Org/rBC ratio of 6.8. Similar to LO-OOA$_{base-4}$, MO-OOA$_{base-4}$ increased during daytime with solar radiation (Fig. 8 and 3b), indicating the important contribution of local photochemistry to MO-OOA production. However, the PCSF plot of MO-OOA also highlighted the significant regional transported contribution, especially from the first three days of this field campaign (Fig. 8, i.e., the data shown in yellow and red color scale for MO-

OOA$_{base-4}$). The MO-OOA $_{base-4}$ factor has a much higher value of $f_{C_2H_4O_2^+}$ (1.2%) than the threshold value of 0.3% derived by Cubison et al. (2011). Nevertheless, the $f_{C_2H_4O_2^+}$ values of MO-OOA $_{base-4}$ reduced to ~0.5% when the conversion factor of 2.45 derived by Ma et al. (2021) is applied for more direct comparison with the standard AMS observation (TV scheme) reported by Cubison et al. (2011). In fact, MO-OOA $_{base-4}$ was strongly associated with K$_3$SO$_4^+$ (R = 0.86), K$^+$ (R = 0.84), and Rb$^+$ (R = 0.68) and gave similar hotspots in their PSCF plots, suggesting that the MO-OOA factor was likely influenced by the

regional transport of biomass burning emissions (Fig. 5 and 8).

PMF$_{base-5}$ solution was investigated to explore whether BBOA can be separated from the existing factors. PMF$_{base-5}$ yields an additional OA factor with the $f_{C_2H_4O_2^+}$ value of 1.6% (denoted as BBOA-HOA$_{base-5}$ in Fig. S7). This new factor increased during nighttime and peaked at ~1:00 LT. The result suggests that PMF$_{base-5}$ attempts to separate the relatively fresh BBOA

components from the total OA, but the origin of this OA factor remains hard to interpret without additional information as its mass spectrum was dominated by fragments similar to those observed in HOA (i.e., C$_x$H$_y^+$, see HOA$_{base-5}$ and BBOA-HOA$_{base-5}$ in Fig. S7). Furthermore, the $f_{C_2H_4O_2^+}$ value of the MO-OOA$_{base-5}$ factor remains high (1.0%), indicating that significant amount of the highly oxidized OA materials still cannot be separated from the relatively fresh BBOA components during the measurement period as they might mix well in the regional air mass before reaching the measurement site.

### 3.4 Improved identification of industrial and biomass burning sources based on PMF$_{metal}$

The six metal-related ions (i.e., Na$^+$, K$^+$, Rb$^+$, V$^+$, Ni$^+$ and K$_3$SO$_4^+$) were integrated into the PMF analysis. PMF$_{base-4}$ and PMF$_{metal-4}$ identify the same types of rBC-containing particles (i.e., rBC-rich, HOA, LO-OOA, and MO-OOA), and there are good correlations of mass spectra and temporal variations between the corresponding factors (i.e., the correlation coefficients

between their mass spectra, R$_{m/z}$ > 0.98, and time series, R$_{ts}$ > 0.94 of PMF$_{base-4}$ and PMF$_{metal-4}$, Table S3). This suggests that addition of metal ions does not significantly affect the performance of PMF analysis for identifying the four major particle types. Figure 10 shows the distribution of the metal ions in each factor from PMF$_{metal-4}$. Majority of K$^+$, K$_3$SO$_4^+$ and Rb$^+$ signals intensities (~85%) were associated with MO-OOA$_{metal-4}$, which provides evidence to support our hypothesis that the MO-OOA$_{base-4}$ factor could be significantly influenced by regional transported biomass burning emissions (see Section 3.3.2). On

the other hand, a significant portion of Na$^+$, Ni$^+$ and V$^+$ signals were apportioned to the HOA$_{metal-4}$ and rBC-rich$_{metal-4}$ factors, highlighting that these two primary factors could be impacted by combustion emissions from shipping and/or industry nearby as previously discussed (see Section 3.2.1).



In this section, the higher number of PMF$_{metal}$ solutions were investigated to improve our understanding of 1) the potential impacts of shipping/industrial emissions on the rBC-rich and HOA factors, and 2) the contributions of biomass burning emissions to MO-OOA due to the regional transport. The PMF$_{metal-7}$ is considered as the optimum solution in this work with
5   the details shown in Fig. 11. Three additional types of rBC-containing particles with reasonable physical meanings, namely industrial and shipping related factor (IOA), aged BBOA (A-BBOA), and BBOA, can be resolved by PMF$_{metal-7}$. The rBC-rich, HOA, LO-OOA and MO-OOA factors identified by PMF$_{metal-7}$ were still the major contributors to rBC and OA masses (Fig. 9), and they gave similar mass spectra (i.e., rBC and OA fragments) and time series compared to those corresponding factors determined by PMF$_{metal-4}$ (R$_{m/z}$ > 0.84, and R$_{ts}$ > 0.92, Table S4), suggesting that the chemical characteristics and
10   temporal variabilities of the four original factors were not significantly affected by the new factors.

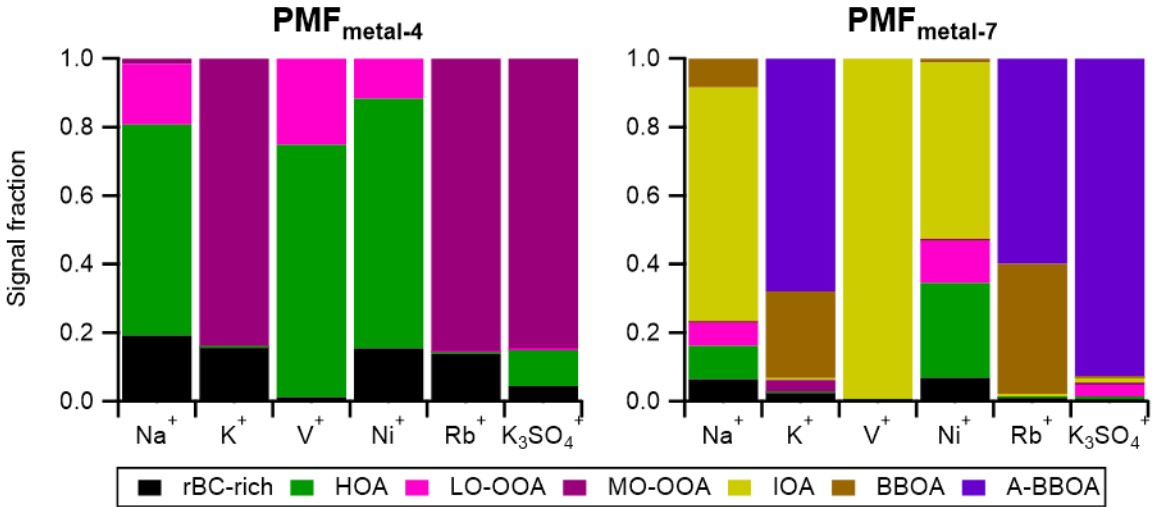

Figure 10. Metal signal contributions to individual factor identified by PMF$_{metal-4}$ (left panel) and PMF$_{metal-7}$ (right panel)







Figure 11. Results of PMF$_{metal-7}$ factors: Mass spectra of OA, rBC and metal (1st column), diurnal cycles of mass concentrations (2nd column, median: dashed lines, 25th/75th percentiles: shaded areas), NWR plots (3rd column), and PSCF plots (4th column). PMF mass spectrum were not RIE adjusted for different species. K$^+$ signal intensities were downscaled by a factor of 10 for visualization purposes. The wind speed unit in NWR plots (radial axis) is m/s. The color scale of NWR plots represents species





concentration ($\mu g/m^3$). Some of the NWR plots presented with different wind speed ranges (< 3 m/s) and resolution are shown in Fig. S8.

### 3.4.1    Identification of rBC-containing particles from industrial emissions

The $IOA_{metal-7}$ factor likely represents primary combustion emissions. This factor is composed of 51wt% of rBC and has organic mass spectra dominated by hydrocarbon fragments (i.e., $C_xH_y^+$). Unlike the $rBC\text{-rich}_{metal-7}$ and $HOA_{metal-7}$ factors, $IOA_{metal-7}$ did not increase during the traffic rush hours (Fig. 11). The concentrations of $IOA_{metal-7}$ substantially increased between 15:00 and 20:00 LT, and the large difference between the median and 75th percentile of $IOA_{metal-7}$ concentrations during the peak period suggests that a very strong IOA emission was occasionally transported from the industrial region and shipping ports to the site

due to prevailing wind. The industrial origin of $IOA_{metal-7}$ is further supported by the large contribution of $Na^+$ (60%), $Ni^+$ (52%), and $V^+$ (~100%) signals to this factor (Fig. 10). By assuming the same RIE for the metal ions, the $V^+/Ni^+$ ratio of the $IOA_{metal-7}$ factor is 1.92.

The $C_1^+/C_3^+$ ratios are similar between $IOA_{metal-7}$ (0.75) and $rBC\text{-rich}_{metal-7}$ (0.73). Although $CO_2^+$ fragments can be originated

from OA and rBC, $CO_2^+$ observed in $rBC\text{-rich}_{metal-7}$ and $IOA_{metal-7}$ are likely largely contributed by refractory soot structure. Both $CO_2^+/C_3^+$ and $C_3O_2^+/C_3^+$ ratios of $IOA_{metal-7}$ (1.39 and 0.11) are higher than that of $rBC\text{-rich}_{metal-7}$ (0.87 and 0.094), suggesting that more oxygenated refractory soot structures of rBC-containing particles were emitted from industry and shipping emissions. As both $IOA_{metal-7}$ and $rBC\text{-rich}_{metal-7}$ are rBC-dominated factors, the Org/rBC ratios of these two factors were estimated by assuming all signals from $CO^+$, $CO_2^+$ and $C_3O_2^+$ contributing to rBC mass ($IOA_{metal-7}$ ~0.2 and $rBC\text{-rich}_{metal-}$

$_7$ ~0.1). The results indicate that the rBC-dominated particles emitted from industrial ($IOA_{metal-7}$) had thicker POA coatings than those associated with traffic ($rBC\text{-rich}_{metal-7}$), although the differences were not substantial. The size distribution of m/z 23 ($Na^+$) signals peaked at ~300 nm during the period with strong influences from industrial emissions (Fig. 7c), suggesting that the rBC-containing particles in primary industrial emissions may have different size characteristics compared to those emitted from local traffic, which gave rBC peaks at ~ 180 nm as reported in Section 3.2.4.

The IOA factor could be resolved starting from $PMF_{metal-5}$, and the temporal variations and mass spectral characteristics remained roughly the same for the IOA factors identified up to the 7-factor solution. By comparing the time series of the rBC-rich and HOA factors determined by $PMF_{metal-4}$ and $PMF_{metal-7}$, $PMF_{metal-7}$ leads to ~16% and 27% of rBC and Org reduction in the mass concentrations of rBC-rich factor, respectively. Reductions in rBC and Org mass concentrations for $HOA_{metal-7}$ were

~48% and 19%, respectively, compared to $HOA_{metal-4}$. The differences of rBC and Org mass concentrations between the sum of rBC-rich and HOA factors (i.e., $rBC\text{-rich}_{metal-7}$ + $HOA_{metal-7}$ vs. $rBC\text{-rich}_{metal-4}$ + $HOA_{metal-4}$ with $R_{ts}$ = 0.97 and 0.98 for rBC and OA masses, respectively) were ~20% and 22%, respectively. Although it is not straightforward to interpret the splitting of



factors from $PMF_{metal-4}$ to $PMF_{metal-7}$, most of the $IOA_{metal-7}$ mass was likely extracted from the rBC-rich and HOA factors. This speculation is supported by our observation that the rBC and Org mass concentrations of these primary factors from $PMF_{metal-4}$ and $PMF_{metal-7}$ are strongly correlated (i.e., rBC-rich $_{metal-7}$ + $HOA_{metal-7}$ + $IOA_{metal-7}$ vs. rBC-rich $_{metal-4}$ + $HOA_{metal-4}$ with $R_{ts} >$ 0.99), and are comparable between the two PMF scenarios with only ~1.5% and 6.9% differences for rBC and Org mass,

respectively.

Our analysis demonstrates that $Na^+$, $Ni^+$, and $V^+$ play a key role to drive the separation of industrial-related combustion emissions for rBC-containing particles from traffic emissions in urban environments that cannot be achieved by $PMF_{base}$ due to non-distinctive rBC and/or OA mass spectral characteristics between local combustion sources. The total rBC mass was

largely assigned to rBC-rich$_{metal-7}$ (71%), followed by $IOA_{metal-7}$ (10%) and $HOA_{metal-7}$ (4%), and these primary factors accounted for ~44% of total OA mass ($HOA_{metal-7}$: 26%, rBC-rich$_{metal-7}$: 13%, and $IOA_{metal-7}$: 5%) on average for the whole period of study as shown in Fig. 9. Using the mean values of their diurnal patterns, it can be estimated that $IOA_{metal-7}$ contributed to up to ~20% of the total rBC and OA mass emitted from the three anthropogenic primary combustion sources during the period with strong industrial influences in the afternoon (i.e., ~80% from rBC-rich $_{metal-7}$ and $HOA_{metal-7}$), whereas the

background contribution of $IOA_{metal-7}$ varied between 8 and 13%. It is worth noting that industrial influences might not be completely separated from the rBC-rich $_{metal-7}$ and $HOA_{metal-7}$ factors in this study due to the similar mass spectral characteristics of rBC and OA emitted from fossil fuel combustion. Therefore, the mass contribution of $IOA_{metal-7}$ probably represents the lower limit at our measurement site. Excluding the period with high IOA mass loadings (i.e., top 25%) in our $PMF_{metal}$, IOA could not be identified and the rBC-rich and HOA mass loadings in the 6-factor solution reduced by 15 and 25% at 17:00-

18:00 LT, respectively.

### 3.4.2   Identification of rBC-containing particles from transported BB emissions

$PMF_{metal-7}$ identified two biomass burning-related factors (BBOA and A-BBOA). $BBOA_{metal-7}$ represents the relatively fresh biomass burning emission as it has strong signal contribution from the two signature fragments (i.e., $f_{C_2H_4O_2^+}$ = 3.8% and

$f_{C_3H_5O_2^+}$ = 1.9%). The $BBOA_{metal-7}$ factor is the second largest contributor of $K^+$ (25%) and $Rb^+$ (38%) signals (Fig. 10). On the contrary, $A\text{-}BBOA_{metal-7}$ likely represents the relatively aged BBOA. A-BBOA $_{metal-7}$ contributed to over 60% of $K^+$ and $Rb^+$, and accounted for 93% of $K_3SO_4^+$. As discussed in Section 3.2.1, $K_3SO_4^+$ has been used as a signature ion for aged biomass burning emissions due to heterogeneous conversion of potassium salt to $K_2SO_4$. During atmospheric aging, OA in biomass burning emissions can be oxidized to diminish the signature fragments (i.e., $C_2H_4O_2^+$) and to produce highly oxygenated OA

that can be indicated by the $CO_2^+$ organic fragments (Cubison et al., 2011). This phenomenon is well aligned with our observation that $A\text{-}BBOA_{metal-7}$ consists of more highly oxidized OA (i.e., O:C = 1.11 and strong signals at $CO_2^+$) and a lower value of $f_{C_2H_4O_2^+}$ (0.9%) compared to those of $BBOA_{metal-7}$. As illustrated in Fig. 12, the $f_{C_2H_4O_2^+}$ values for $BBOA_{metal-7}$ and $A\text{-}BBOA_{metal-7}$ became 1.5% and 0.3%, respectively with the conversion factor applied (Ma et al., 2021). The data without




applying the conversion factor is shown in Fig. S9. The hotspots of BBOA$_{metal-7}$ and A-BBOA$_{metal-7}$ in the PSCF plots suggest the regional transport of these two types of rBC-containing particles from northeast of Singapore (Fig. 11).

The contribution of LO-OOA and MO-OOA derived from PMF$_{metal-7}$ to total OA mass reduced by 5% and 7%, respectively,
compared to the case of PMF$_{based-4}$ (Fig. 9). As discussed in Section 3.3.2, MO-OOA$_{base-4}$ represents the more oxidized SOA that was influenced by both local photochemistry and regional transport of biomass burning emissions, whereas LO-OOA$_{base-4}$ is primarily associated with local photochemical production. By comparing the results between PMF$_{metal-4}$ and PMF$_{metal-7}$, PMF$_{metal-7}$ leads to reduction in ~58% and 23% of rBC and OA mass concentrations, respectively, for the total of these two OOA factors (i.e., LO-OOA$_{metal-7}$ + MO-OOA$_{metal-7}$ vs. LO-OOA$_{metal-4}$ + MO-OOA$_{metal-4}$ with R$_{ts}$ = 0.93 and 0.97 for rBC and
OA masses, respectively), which were mainly redistributed to the BBOA$_{metal-7}$ and A-BBOA$_{metal-7}$ factors. There were only within less than 2% differences on average between the total rBC and OA mass concentrations of these factors derived from PMF$_{metal-4}$ and PMF$_{metal-7}$ (i.e. LO-OOA$_{metal-7}$ + MO-OOA$_{metal-7}$ + BBOA$_{metal-7}$ + A-BBOA$_{metal-7}$ vs. LO-OOA$_{metal-4}$ + MO-OOA$_{metal-4}$ with R$_{ts}$ = 0.98 and > 0.99 for rBC and OA masses, respectively). The above results show that PMF$_{metal}$ can allow better separation of aged/more oxidized biomass burning-related rBC-containing particles from locally formed and background
SOA materials at our measurement site. This is further supported by our observation that the $f_{C_2H_4O_2^+}$ values of MO-OOA$_{metal-7}$ substantially dropped to 0.1% compared to that of MO-OOA$_{base-4}$ (0.5%, Fig. 12).

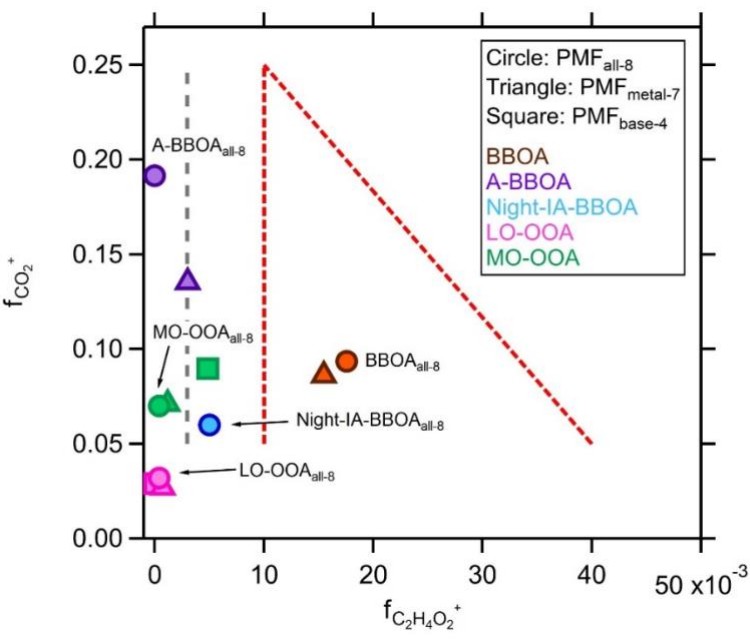



Figure 12: The $f_{CO_2^+}$ vs. $f_{C_2H_4O_2^+}$ plot for evaluating OOA and biomass burning-related OA factors derived from different PMF scenarios. The conversion factor of 2.45 derived by Ma et al. (2021) was applied to the $f_{C_2H_4O_2^+}$ values for more direct comparison with the observational-based framework derived by Cubison et al. (2011). The data without conversion was reported in Fig. S9. Grey dashed line represents background level (0.3%) of $f_{C_2H_4O_2^+}$ for OA without significant biomass burning signature reported by Cubison et al. (2011). Red dashed lines represent the boundaries of triangular region that cover BBOA observed by Cubison et al. (2011).

## 3.5  Additional insight into SOA chemistry and aging of POA based on PMF$_{all}$

Previous studies have shown that including IA species related to secondary processes in PMF analysis of standard HR-AMS measurements can provide further insights on source separation and characteristics (Sun et al., 2012; Xu et al., 2015; Ng et al., 2017; Day et al., 2022). This section focuses on integrating a few IA components, including fragments from $SO_4^{2-}$, $NO_3^-$ and $Cl^-$, into the PMF$_{metal}$ to explore whether the understanding of secondary aerosol coating formation and aging chemistry of rBC-containing particles can be improved. Similar to PMF$_{base-4}$, PMF$_{all-4}$ yields the rBC-rich, HOA, LO-OOA and MO-OOA factors, and all of them correlates well with the corresponding PMF$_{base-4}$ factors ($R_{m/z} > 0.98$ and $R_{ts} > 0.98$, Table S3), suggesting high consistency between all the 4-factors PMF scenarios (i.e., PMF$_{base-4}$, PMF$_{metal-4}$ and PMF$_{all-4}$). The physical meaning of higher number factors for PMF$_{all}$ was explored, and PMF$_{all-8}$ is considered as the optimum solution. These four basic factors identified by PMF$_{all-4}$ also gave similar time series compared to those corresponding factors determined by PMF$_{all-8}$ ($R_{ts} > 0.94$, Table S5).

PMF$_{all-8}$ was also able to identify the same seven factors as PMF$_{metal-7}$, and to further separate a biomass burning-influenced OA factor that were associated with different IA fragments (denoted as Night-IA-BBOA$_{all-8}$ hereafter). Integrating IA fragments into the PMF analysis did not make significant impact on the distribution of metal ions in fossil and biomass combustion emissions (Fig. 13). IOA$_{all-8}$ is still the largest contributors to $Na^+$, $V^+$, and $Ni^+$. BBOA$_{all-8}$ and A-BBOA$_{all-8}$ remain the two main contributors to $K^+$, $Rb^+$, and $K_3SO_4^+$ (Fig. 10 and 13). According to the $CO_2^+/C_3^+$ and $C_3O_2^+/C_3^+$ ratios, the IOA$_{all-8}$ factor (1.23 and 0.10) contained more oxygenated refractory soot structures compared to that associated with the rBC-rich$_{all-8}$ factor (0.82 and 0.09), which is also similar to the results of PMF$_{metal-7}$ (Section 3.4.1). As reported in Table S2, the Org/BC ratios of rBC-rich, IOA, and LO-OOA between PMF$_{metal-7}$ and PMF$_{all-8}$ remains roughly the same, whereas the ratios of HOA, MO-OOA and BBOA decreased or increased by 23-32%. The Org/rBC ratio of A-BBOA$_{all-8}$ is about 2.5 times higher than that of A-BBOA$_{metal-7}$. Overall, most of the rBC mass (~78%, i.e., rBC-rich$_{all-8}$ + IOA$_{all-8}$) was thinly coated by OA (i.e., Org/rBC < 1), and only ~4% of rBC mass was thickly coated by HOA$_{all-8}$ and LO-OOA$_{all-8}$ materials (Org/rBC > 13.9). rBC mass associated with MO-OOA$_{all-8}$ and all the BBOA-related factors identified by PMF$_{all-8}$ (~18%) was moderately coated by OA (Org/rBC = 2.3-7.4).





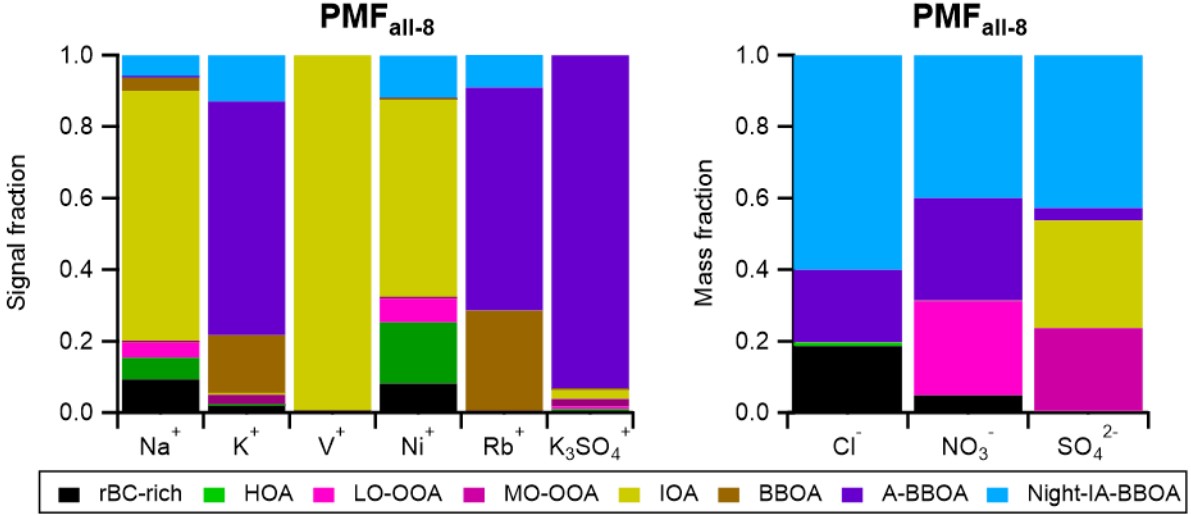

Figure 13. Metal (left panel) and IA components (right panel) contributions to individual factor identified by PMF$_{all-8}$

**3.5.1    Identification of BBOA associated with IA chemistry.**

Night-IA-BBOA$_{all-8}$ accounted for 7% and 9% of total rBC and OA mass, respectively (Fig. 7). Figure 13 shows that 6%, 13%, 12%, and 9% of Na$^+$, K$^+$, Ni$^+$ and Rb$^+$, respectively, are allocated to the Night-IA-BBOA$_{all-8}$ factor, suggesting that this factor could have experienced mixed influences from both industrial and biomass burning sources. Furthermore, the major IA fragments in BBOA$_{all-7}$ (48%, 39%, and 63% of SO$_4^{2-}$, NO$_3^-$, and Cl$^-$, respectively) were largely redistributed to Night-IA-

BBOA$_{all-8}$ (43%, 40%, and 60% of SO$_4^{2-}$, NO$_3^-$, and Cl$^-$, respectively). Together with the diurnal pattern of Night-IA-BBOA$_{all-8}$ (Fig. 14), our observation suggests that Night-IA-BBOA$_{all-8}$ was more associated with the formation and/or condensation of secondary IA at night before they reached the sampling site. A-BBOA$_{all-8}$ is another factor that had observable contributions to the three major inorganic components (4%, 29%, and 20% of SO$_4^{2-}$, NO$_3^-$, and Cl$^-$, respectively).

The mixed influences from multiple emissions and aging processes were also reflected by the relatively even reduction in OA masses of the rBC-rich, MO-OOA and BBOA-related factors (~3% each) compared to the results of PMF$_{metal-7}$ (Fig. 9). Comparing to MO-OOA$_{metal-7}$, separation of Night-IA-BBOA$_{all-8}$ from other OOA components led to ~28% of reduction in MO-OOA$_{all-8}$ background mass concentrations between midnight and early morning (i.e. from ~0.06 to 0.04 µg/m$^3$). In contrast, no obvious changes in background levels of LO-OOA were observed between the PMF$_{metal-7}$ and PMF$_{all-8}$ solutions (i.e., 0.05

µg/m$^3$ for both cases). In addition to the contribution of Na$^+$ and Ni$^+$, clear mass spectral pattern of hydrocarbon peaks (i.e., C$_x$H$_y$) was observed for Night-IA-BBOA$_{all-8}$, and was similar to that observed in HOA$_{all-8}$ and rBC-rich$_{all-8}$. The relatively high



H:C ratio (1.66) and low $OS_c$ (-0.26) of Night-IA-BBOA$_{all-8}$ compared to other PMF$_{all-8}$ factors also indicates the contribution of POA from combustion emissions to this nighttime factor. These observations illustrate the possible involvement of fossil fuel combustion related POA in nighttime chemistry although their emissions were relatively low at midnight in general. The $f_{C_2H_4O_2^+}$ values of Night-IA-BBOA$_{all-8}$ is 0.5% (or 1.3% without the conversion factor applied). This BBOA signature fragment

is still slightly higher than the background levels for typical POA and SOA (Cubison et al., 2011) although they might have experienced extensive mixing with other aerosol components and/or atmospheric processing at nighttime (Fig. 12 and S9).

BBOA$_{all-8}$ retained the signature organic fragments of biomass burning emissions ($f_{C_2H_4O_2^+}$ = 1.8% or 4.3% without conversion factor applied), whereas almost no $f_{C_2H_4O_2^+}$ was observed in A-BBOA$_{all-8}$ likely due to the improved separation of more aged

BBOA components from the fresher biomass burning emissions in PMF$_{all-8}$ solution. Nevertheless, A-BBOA $_{all-8}$ remained the most important contributor to K$^+$, Rb$^+$, and K$_3$SO$_4$$^+$, which continue to support our hypothesis that A-BBOA likely represents the heavily aged OA in biomass burning emissions. These observations further highlight the importance of including metal-related ions for PMF analysis of OA coated on rBC core for source apportionment analysis. The origin of heavily aged/oxidized biomass burning-related OA materials (e.g., A-BBOA identified in this work) cannot be easily identified based on the

observational-based framework (f$_{44}$  vs. f$_{60}$ or $f_{CO_2^+}$ vs. $f_{C_2H_4O_2^+}$) developed  by Cubison et al. (2011) alone as the signature BBOA fragments can be completely diminished after extensive aging. Such additional information is required to better understand the atmospheric fate and transformation of OA in biomass burning emissions and how they may affect their contribution to the highly oxidized background SOA materials in various locations as demonstrated in this work.





Figure 14. Results of PMF$_{all-8}$ factors: Mass spectra of OA, rBC, metal, and IA (1st column), diurnal cycles of mass concentrations (2nd column, median: dashed lines, 25th/75th percentiles: shaded areas), NWR plots (3rd column), and PSCF plots (4th column). PMF mass spectrum were not RIE adjusted for different species. K$^+$ signal intensities were downscaled by





a factor of 10 for visualization purposes. The wind speed unit in NWR plots (radial axis) is m/s. The color scale of NWR plots represents species concentration ($\mu g/m^3$). Some of the NWR plots presented in different wind speed ranges (< 3 m/s) and resolution are shown in Fig. S10.

### 3.5.2    Insight from nitrate and sulfate fragments

Figure 13 shows that $NO_3^-$ mass was mainly assigned to Night-IA-BBOA$_{all-8}$ (40%), A-BBOA$_{all-8}$ (29%), and LO-OOA$_{all-8}$ (26%). As discussed in Section 3.2.3, the $NO^+/NO_2^+$ ratio can be used as an indicator to evaluate the relative importance of organo-nitrate and inorganic nitrate. The $NO^+/NO_2^+$ ratios for LO-OOA$_{all-8}$, A-BBOA $_{all-8}$, and Night-IA-BBOA$_{all-8}$, were 0.8, 1.7, and 3.3, respectively. Although the separation of organo-nitrate and inorganic nitrate in NR-PM$_{coating}$ determined by the

LV scheme of SP-AMS remains non-quantitative due to different fragmentation for compounds vaporized by TV and LV (Ma et al., 2021), there are two major implications for these observations. Firstly, the $NO^+/NO_2^+$ ratio of LO-OOA$_{all-8}$, was close to the one for $NH_4NO_3$ coated on rBC cores (0.9) determined during our SP-AMS calibration. As LO-OOA $_{all-8}$ mainly represents the freshly formed SOA due to local photooxidation, our results suggest that inorganic nitrate and fresh SOA with lower degree of oxygenation (i.e., LO-OOA $_{all-8}$) were likely produced together via photooxidation of $NO_2$ and gaseous SOA precursors, and

they subsequently condensed on the rBC particles in a local time scale. Secondly, the relatively high $NO^+/NO_2^+$ ratio of the A-BBOA $_{all-8}$ and Night-IA-BBOA$_{all-8}$ factor indicates that organo-nitrate compounds can be significant contributors to the observed $NO^+$ and $NO_2^+$ signals in these factors. This is well aligned with recent laboratory studies that nocturnal $NO_3$ radical chemistry of different SOA precursors, including those emitted from biomass burning, can be a secondary source of organo-nitrate compounds (Fry et al., 2018; Kodros et al., 2020; Yazdani et al., 2023).

The majority of $SO_4^{2-}$ mass was assigned to Night-IA-BBOA$_{all-8}$ (42%), IOA$_{all-8}$ (30%), and MO-OOA$_{all-8}$ (23%) as shown in Fig. 13. There are two implications from this observation that may require further investigation. Firstly, although IOA $_{all-8}$ is expected to be primary in nature, the relatively large contribution of this factor to $SO_4^{2-}$ mass suggests that the rBC-containing particles from industry may act as a condensation sink of acidic sulfate that are locally formed through photooxidation of

gaseous $SO_2$ in the pollutant plumes emitted from industry as discussed in Section 3.2.3. Our results show that LO-OOA and MO-OOA represent freshly formed SOA due to the strong solar radiation in the afternoon in all PMF cases. PMF$_{all-8}$ further suggests that formation of LO-OOA$_{all-8}$ and MO-OOA$_{all-8}$ were likely concurrent with nitrate and sulfate formation chemistry, respectively, in daytime. The formation of these two types of OOA particles might be due to the mixing of different gaseous precursors in the pollutant plumes near sources and during atmospheric transport. Secondly, in general, our findings show that

both A-BBOA$_{all-8}$ and Night-IA-BBOA$_{all-8}$ were the key contributors of $Cl^-$ and $NO_3^-$ components, whereas there was a large difference of $SO_4^{2-}$ mass contribution between the two factors. Despite the fact that $SO_4^{2-}$ contribution from A-BBOA$_{all-8}$ was relatively small, A-BBOA$_{all-8}$ accounted for 93% of $K_3SO_4^+$ signals, indicating the association of $SO_4^{2-}$ formation chemistry



to this factor. As K$_2$SO$_4$ can be formed via reactions of acidic sulfate with other potassium salts (e.g., KCl) in biomass burning emissions, these observations imply that different forms of sulfate fragments measured by the SP-AMS may be able to evaluate atmospheric aging of biomass burning emissions involving interaction with acidic sulfate. Overall, our observations highlight the importance of studying the interaction of SOA and IA formation chemistry and aerosol mixing state in such complex urban
environments.

## 4.    Conclusion and atmospheric implications

With an increasing number of SP-AMS measurements reported around the world (Zhang et al., 2020), source apportionment of rBC-containing particles and their associated coating materials based on PMF analysis can improve the understanding of
characteristics and atmospheric processing of various combustion emissions. While the current common practice is either using OA fragments alone or the combination of OA and rBC fragments as the input of PMF analysis, this work demonstrates the importance of performing PMF analysis of all the chemical species that can be measured by SP-AMS. Including rBC, OA, IA, and metals fragments improve source apportionment and coating thickness quantification (i.e., Org/rBC ratio) of rBC-containing particles in urban environments under strong influences of multiple anthropogenic emissions, and provide insight
into our understanding of secondary processing of rBC-containing particles in the atmosphere.

This work clearly illustrates that including metal signals in the PMF analysis (PMF$_{metal}$) can significantly improve the quality of source apportionment of rBC-containing particles in Singapore, a complex tropical urban environment with large scale shipping ports and petrochemical industries. Compared to the PMF$_{base}$ that can only identify two primary factors (HOA and
rBC-rich OA) associated with different rBC content and coating thickness mainly emitted from local traffic, PMF$_{metal}$ and PMF$_{all}$ can further separate another primary source from industry and shipping emission. The IOA factor was strongly associated with heavy metals (e.g., V$^+$ and Ni$^+$), suggesting that this type of rBC-containing particles might pose higher potential risks to human health. Although such industrial and shipping emissions are transported to our sampling occasionally during sea breeze in the afternoon, the IOA factor accounted for 10-12% of rBC and and 5% of OA coating, indicating that
the negative impacts of industrial emissions on the local air quality cannot be ignored for the development of air pollution mitigation strategy.

Integrating metal and IA components into PMF analysis can provide even further insight into understanding the fate and transformation of rBC-containing particles in aged biomass burning emissions, and the origin of heavily aged SOA in the
background. Two BBOA factors with different degree of oxygenation (BBOA and A-BBOA) were identified by PMF$_{metal}$ and PMF$_{all}$, accounting for 6-9% and 13-16% of rBC and OA coating mass in total, respectively. A-BBOA accounted for over 95% of K$_3$SO$_4^+$ signals that could be due to heterogeneous production of potassium sulfate in biomass burning plumes. Recently, Vasilakopoulou et al. (2023) have shown rapid transformation of BBOA into highly oxidized and harmful background MO-

OOA in Europe. Although the mass contribution of A-BBOA factor was relatively small in this work, our results illustrate the possibility of separating highly oxidized BBOA coating materials from the typical background MO-OOA to better evaluate the potential impacts of aged BBOA to human health and climate. The associations between $PMF_{all}$ can further identify the Night-IA-BBOA factor, representing aged-BBOA component that was strongly associated with nighttime IA and organo-

nitrate formation. Overall, A-BBOA and Night-IA-BBOA derived from $PMF_{all}$ accounted for 12% of OA coatings mass in total, and the mass contribution of MO-OOA to total OA coatings decreased from 31% for $PMF_{base}$ to 24 and 21% for $PMF_{metal}$ and $PMF_{all}$, respectively.

Most of the IA species observed in this work were externally mixed rBC particles (i.e., > 95% IA were externally mixed).

However, including the IA coating materials in the PMF analysis still can provide additional information to advance our understanding of SOA coating formation. $PMF_{all}$ shows concurrent LO-OOA and inorganic nitrate formation via daytime chemistry, suggesting that the LO-OOA formation observed in this work is likely more relevant to the photochemical SOA formation chemistry in the presence of $NO_x$. On the other hands, the observed linkage between local photochemical production of MO-OOA and acidic sulfate particle formation suggests the possibility of co-oxidation of $SO_2$ and SOA precursors in the

industrial-influenced atmosphere. These observations highlight the importance of investigating the interaction of SOA and IA formation chemistry and integrating such details in air quality models, and illustrates the need of evaluating the degree of aerosol mixing of rBC and chemical species in NR-$PM_{coating}$ as well as its relationship with different atmospheric processes by utilizing the single particle measurement capability of SP-AMS (Willis et al., 2016; Lee et al., 2016; Lee et al., 2019; Farley et al., 2024).

### Data availability

The dataset for this publication is available upon contacting the corresponding authors.

### Author contribution:

AKYL supervised the projects and conceptualized the research goals. MM, LHR and AKYL planned the field campaign and performed data analysis. MM and LHR, carried out SP-AMS measurements. YZ, MK, and LEY provided supporting measurement data. MM and AKYL prepared the manuscript with input from all the co-authors.

### Competing interests:

AKYL is the member of the editorial board of Atmospheric Chemistry and Physics.





## Acknowledgements

This work was supported by the National Environmental Agency of Singapore (NEA; grant no. R-706-000-043-490), Environment and Climate Change Canada Air quality program and Cambridge Centre for Carbon Reduction in Chemical Technologies program. The contents of this paper do not represent the views of the National Environmental Agency of Singapore.

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
