# Peer review of "Advances in characterization of black carbon particles and their associated coatings using the soot particle aerosol mass spectrometer in Singapore, a complex city environment"

_EGUsphere, 2024_

## Author Comment (AC1)

**Response to the reviewer comment 1:**

This manuscript describes analysis of soot particle aerosol mass spectrometer (SP-AMS) measurements conducted in Singapore during a ~2-month period. The manuscript systematically explores and demonstrates the application of PMF to the SP-AMS data. The analysis starts with a more traditional approach of using organic aerosol (and refractory BC, rBC) signals and then sequentially adds in metal signals and then inorganic signals, all from the SP-AMS spectrum. With the traditional approach, 4 meaningful source factors are resolved. With the addition of the metals, another 3 factors are resolved, an aged biomass burning and a fresher primary biomass burning factor, as well as an industrial and shipping related factor. Addition of the inorganic signals yield one more factor, a biomass burning-influenced OA factor associated with different inorganic fragments. Various additional formulations are used to provide support for the meaningfulness and robustness of the PMF separations, e.g. size diurnal cycles, size distributions, wind and back trajectory analyses, as is typically required for testing PMF results. Apportionment of the rBC among the factors and the contributions of the factors to overall composition are reported. The manuscript is primarily a methods demonstration, with a few relatively generalized observations of aerosol chemical and source processes.

The paper is well-organized and clearly written. It will make a valuable contribution to the body of literature of aerosol composition analytical methods. I liked the way the expansion/exploration of the additional PMF chemical inputs were discussed sequentially, building on the previous results. I also particularly found Fig 12 to be a valuable in putting these new and previous BBOA factor separations in context. I recommend publication in EGUsphere with a few minor suggested changes.

Response: We thank for the positive feedback from the reviewer on the significance of this work. Our responses to specific comments are in blue color below:

A few general comments. While I understand that the primary focus of this work is methods demonstration, it would seem useful to provide more of a general presentation of previous aerosol measurements conducted in this region for context. There are a few statements and references that are sprinkled in at points in the manuscript to compare to relevant observations. However, it was difficult as a reader not familiar with this region to get a sense of what is known about aerosol concentration, composition, and sources in this region. Also, during particularly the 1st half of the manuscript, there were a number of instances where the discussions and interpretations were a bit overreaching or protracted discussion of details that may not add any additional information. Those probably at unnecessarily length to an already fairly long manuscript and water down the more interesting and insightful information in this manuscript. See for example the detailed comments below on nitrate composition (P16, L25), diurnal cycles of O/C (P15, L8-11), and ions fragment size distributions (P17, L25-30). I encourage the authors to review the paper and trim any unnecessary text and discussions along those lines.

Response:

We agree with the reviewer's comment that the reader can benefit by including more information about the PM concentrations and their potential large sources in our sampling location. The co-located ACSM and aethalometer measurements can provide more details of NR-PM$_1$ and BC during the sampling period of this work, and the relevant discussion has been included in the revised manuscript. The details is provided under the response for the specific comments from both reviewers. As suggested by the reviewer, some of the discussions have been shortened or moved to supplementary information.

In addition, some minor modifications have been made in the last paragraph of the introduction to improve the description of the complexity of PM sources in the urban area of Singapore as shown below.

Page 4 Lines 14-21: Singapore is a highly developed coastal urban environment located at the end of the Malay Peninsula in Southeast Asia. Aside from typical engine emissions from local traffic and local cooking emissions (Budisulistiorini et al., 2018; Rivellini et al., 2020), Singapore has the world's second busiest shipping port (World Shipping Council, 2024) and fifth largest oil refining capacity (Lau et al., 2021), which can be potential large anthropogenic combustion sources of air pollutants. Furthermore, together with atmospheric processing during transport, air quality and concentrations of atmospheric PM in Singapore can be significantly influenced by SOA (Budisulistiorini et al., 2018; Rivellini et al., 2020) and transboundary haze due to regional/transported biomass burning emissions in Southeast Asia (Budisulistiorini et al., 2018; Heil and Goldammer, 2001; Kuwata, 2024).

Detailed Comments:

P7, L8-9. Incomplete sentence

Response:

The manuscript has been revised as shown below:

Page 7 Lines 10-11: Noting that $mIE_{NO3}$ and $mIE_{rBC}$ were determined when the LV was off and on, respectively. The TV was operated at 600 °C throughout the calibration.

P8, L20-21. Would be useful to explain the methodological reasoning for downweighting $K^+$ and upweighting $Rb^+$, $V^+$, and $Ni^+$. I.e. how does this improved separation, what happens if you keep them at their native S/N? Of if this was explored more systematically in the papers referenced, perhaps briefly state how it can help. These types of details/discussions could be quite useful for others conducting "combined" PMF" as indeed this manuscript is in large part a methods demonstration.

Response:

Downweight and upweight signals of AMS data for PMF analysis has been more systematically investigated by previous studies to improve the separation of factors (Adam Reff et al., 2012; Lee and Hopke, 2006; Paatero and Hopke, 2003). In general, OA fragments with signal-to-noise ratios (S/N) less than 0.2 are "bad" signals, and S/N between 0.2-2 are "weak" signals that require downweight or removal. A general description can be found for the PMF analysis of AMS data (https://cires1.colorado.edu/jimenez-group/wiki/index.php/ToF-AMS_Main), but the decision largely relies on specific data set and user judgment.

As described above, the general goal of signal adjustment is to generate comparable S/N values to a major fragment S/N (e.g., $C_2H_3O^+$ has an S/N ~25, $C_3^+$ has an S/N ~21). $K^+$ had a native S/N of ~130, which was too strong, leading to poor factor separation and requiring downweight. In contrast, $Rb^+$, $V^+$ and $Ni^+$ had S/N below 2 that required upweight to generate meaningful results. Figure R1 shows the comparison of 7-factor solutions using default S/N, adjusted S/N in $PMF_{metal-7}$, and over-adjusted S/N (i.e., S/Ns < 0.2), using metal ions as an example. Default S/N (Fig. R1a) led to a factor dominated by $K^+$ (> 99% signal contribution). Our $PMF_{metal-7}$ had a good factor separation as metal S/Ns were between 2-20. When the S/N were low (i.e., S/N < 0.2, Fig. R1c), the factor separation was not affected by the addition of metal ions.

[Figure]

Figure R1. A 7-factor PMF solutions for metal ion with original and adjusted S/N.

The manuscript has been revised as shown below to highlight that the metals were downweighed or upweighted to generate comparable S/N ration to OA and rBC fragments in our PMF analysis.

Page 8 Lines 23-25: The signal of $K^+$ was downweighed by 10 times, whereas $Rb^+$, $V^+$, and $Ni^+$ were upweighted by 3 times to generate comparable signal-to-noise ratio (S/N) strengths to OA and rBC fragments.

P8, L25-26. "Sulfate fragments were downweighed by a factor of 3 due to their relative strong intensities." For PMF, it's the S/N, not the signal that controls the influence on the factorization. I.e. if intensity (signal) is high but so is the uncertainty, it's not necessarily going to have a strong influence. This is a useful paper, that I think future practitioners will read, so import to be precise with language.

Response:

Following the previous metal S/N comment, we have modified the sentence for the case of sulfate as shown below.

Page 8 Lines 28-29: Sulfate fragments' intensities were downweighed by a factor of 3 due to their relative strong S/N.

P8, L29. "FPEAK" not "F peak"

Response:

This error has been corrected as suggested.

Fig. 2e pie chart. In the time series it appears that Cl is on average greater than SO4, however it is not visible in the pie chart, while SO4 is?

Response:

Chloride mass is ~0.03% of total aerosol mass. The time series of $Cl^-$ mass was scaled up by a factor of 20. The legend of Figure 2e has been updated to present this information.

P11, L10-11: Here quantitatively comparing SP-AMS rBC to aethalometer. It would seem key to discuss the absorption coefficients applied to the AE33 data (to convert optical absorption to mass) for comparison to be most meaningful.

Response:

We agree that the mass absorption coefficient (MAC) is critical to calculate equivalent BC (eBC) concentrations measured by the aethalometer. As aethalometer is more robust compared to SP-AMS for BC measurement, the primary goal of our time series comparison is to demonstrate that rBC mass concentrations measured by the SP-AMS are reasonable and their temporal variations can be captured due to changes in BC source contribution.

It is important to point out that quantification of rBC using SP-AMS strongly depends on the degree of overlapping between LV and particle beam, which ultimately affects the $RIE_{rBC}$ values and collection efficiency of rBC particles (Willis et al., 2016). The coating thickness of rBC particles, which varies significantly within atmospheric BC particle population, can affect the particle beam width and subsequently the rBC quantification. Therefore, using the SP-AMS-measured rBC concentrations to evaluate the accuracy of MAC values applied in the aethalometer may not be appropriate, and such instrument comparison is out of the scope of this work. In this study we use the default MAC value of 7.77 $m^2/g$ to calculate eBC concentration (Drinovec et al., 2015).

The manuscript has been revised as shown below:

Page 11, Lines 11-15: The rBC concentrations measured by the SP-AMS was about 31% higher than the equivalent BC (eBC) concentrations measured by the aethalometer, a strong temporal correlation between the two measurements was observed, suggesting that the temporal variation of BC concentrations can be captured reasonably well by both instruments (R = 0.89, Fig. S4). Such strong correlation between the eBC and rBC concentrations was consistently observed at the same sampling location (Rivellini et al., 2020).

P15, L8-11: O/C values of 0.38-0.39 on average during the morning and evening rush hours is a lot higher than typically HOA O/C ratios (typically <0.1), at least for standard vaporizer AMS. Authors should reconsider this statement or provide context for differences for the SP-AMS (perhaps discussed in more detail later, but this would seem to stand out as written here as a dubious statement for most readers with some familiarity of AMS OA analyses).

Similarly, in P15, L17-19 the higher evening O/C (~0.43) are interpreted as "suggests that the OA coatings were largely contributed by oxidized organic species". Together these statements suggest that a shift from 0.38-0.39 to 0.43 indicates a shift from mostly HOA to mostly OOA coatings. That seems like a big exaggeration of what appears to be a fairly subtle shift between degrees of moderately oxidized composition. Or perhaps I'm misunderstanding the intention, in which case consider making the point more precisely.

Response:

As the reviewer pointed out, the fragmentation of OA by LV of SP-AMS can be different compared to the standard AMS as reported in our previous work regarding the impacts of LV scheme on the elemental analysis of OA coating on BC particles (Ma et al., 2021). Note that the O:C ratios of HOA factor identified in our PMF solutions ranged from 0.1 to 0.14, which are comparable to the typical HOA factor observed from TV scheme of standard AMS measurement. The relatively high O:C ratios during the morning rush hours compared to that of typical HOA could be due to the oxygenated soot functionality in the rBC-rich factor due to traffic emissions. The O:C ratios of LO-OOA and MO-OOA were about 0.5 and 1, respectively as shown in the PMF solutions.

We agree with the reviewer that we may over-interpret some of the results based on the O:C ratio alone in this section. As the PMF results reported in the later sections provide evidence to support our arguments, we have revised/shortened the manuscript as follows.

We have modified the description about the impacts of OOA coating formation on the O:C ratios in this section.

Page 15, Lines 17-19: The higher O:C ratio and $OS_c$ overnight suggests increasing contribution of oxidized organic species to the total OA coatings, which is consistent with greater influence from regional transport of aged aerosols with biomass burning influences overnight, as discussed above in Section 3.2.1.

We have moved the argument about the characteristics of traffic emissions (i.e., HOA) to the later section with the support from PMF results.

Page 17, Line 31 – Page 18, Line 1: Therefore, the observation of lower Org/BC, O:C and $OS_c$ in Fig. 3 indicates that the rBC emitted from local traffic was mainly coated by POA and the thickness of such POA coatings was relatively thin for the majority of the rBC particles compared to those observed during the rest of the day.

P16, L1-7. The authors reference to the Cubison et al background value (0.3%) determined for a standard vaporizer AMS, but then discuss enhancement factors for oxygenated OA and levoglucason, as well as point to the f values in this data (Fig. 6). What are those enhancement factors referenced to? Cubison 0.3% or some revised LV-SP-AMS background? Can the authors report the typical background $fC2H4O2+$ for non-BB OA for the LV-SP-AMS, assuming that's what the factors are being reference to? Adding a horizontal line in Fig. 6d for background f's for LV SP-AMS might be useful as well.

Response:

We have revised the manuscript to deliver a clearer message on the calculation of $f_{C_2H_4O_2^+}$ background level based on the observations from Cubison et al. (2011) and Ma et al. (2021). A dashed line indicating potential $f_{C_2H_4O_2^+}$ background has been added to Fig. 6d along with a proper caption. The manuscript has been revised as shown below:

Page 15, Lines 2-4: Figure 6. Diurnal cycles of (a) $V^+/Ni^+$, (b) $K_3SO_4^+/K^+$, (c) $NH_4^+{}_{meas}$ / $NH_4^+{}_{pred}$, (d) $f_{C_2H_4O_2^+}$ and (e) $NO^+/NO_2^+$ measured by SP-AMS. The horizontal dashed line in panel (d) represents the background level of $f_{C_2H_4O_2^+}$ for non-BBOA reported by Cubison et al. (2011) with the correction factor for OA measured by LV scheme applied (Ma et al., 2021).

Page 15, Line 19 – Page 16, Line 9: To investigate the possible biomass burning influences on OA coatings, a mass contribution of a biomass burning tracer fragment to the total organic coating ($f_{C_2H_4O_2^+}$) was determined as shown in Fig. 6d. Cubison et al. (2011) reported that $f_{C_2H_4O_2^+} > 0.3\%$ obtained from standard AMS measurements

can be the indication of biomass burning influences. Ma et al. (2021) showed that $f_{C_2H_4O_2^+}$ can be enhanced by a factor of ~2.45 on average for oxygenated organic coating and 2.33 for levoglucosan vaporized by the LV scheme of SP-AMS. Therefore, the background level of $f_{C_2H_4O_2^+}$ was re-calculated using the correction factor of 2.33 as shown by the dashed horizontal line in Fig. 6d. Although the $f_{C_2H_4O_2^+}$ values of total OA were close to the background level, the higher values of $f_{C_2H_4O_2^+}$ and $OS_c$ of OA coating materials between 2:00 and 8:00 LT suggests potential influences of biomass burning emissions on the chemical characteristics of rBC-containing particles during the night. The potential contribution of fresh and aged biomass burning emissions will be further discussed based on the PMF analysis (see Section 3.3).

P16, L25. "$NO_3^-$ and $Cl^-$ have elevated levels over midnight…" Does Fig. 6d really support elevated NO3? Nightime NO3 doesn't seem higher than the average values and any diurnal cycles in the average look quite subtle. And given the variably/range (shown in swaths), those features barely seem significant. This appears to be another example of where the authors are reaching to draw conclusions/speculations from very subtle features.

Response:

As the nitrate and chloride concentrations were very low in this study, we have removed the argument about the nitrate and chloride elevated levels over midnight in the revised version to avoid over-interpretation. The current discussion focuses on the diurnal changes in $NO^+/NO_2^+$ ratio.

P17, L25-30: speculation on the m/z 43, 55, 57 peaks being $C_xH_y^+$ or $C_xH_yO_z^+$ peaks. Those peaks can be separated peaks using the HR PToF, assuming the .p files were saved. Otherwise, in this section, discussion of those ions doesn't seem to add any additional information. E.g. the distribution of m/z 44 and laboratory measurements are used to speculate what the composition of the m/z 43 is? So then, what does showing/discussing the measured m/z 43 distribution add?

Response:

We agree with the reviewer that the size distribution may not provide much new information, and they have been modified and moved to SI to reduce the main text length.

Sect 3.3.1 POA from fossil fuel combustion. This is interesting. Can the authors discuss why they think the rBC-rich and HOA factors were separated with PMF (case base-4) and what information that separation can provide? It is noted that both factor sources presumably come from combustion, and the diurnal cycles, NWR and PSCF plots are very similar. But clearly, there must be some significant difference in the time series from some changes in atmospheric sources/processes, instrumental drifts, etc.

Response:

There are two major differences between the temporal variations of HOA and rBC-rich factors. Firstly, the diurnal profiles of the rBC-rich factor peaked one hour earlier than the HOA factor in the morning rush hours. Secondly, although the NWR analysis shows high concentrations of both HOA and rBC-rich factors at the hotspot observed from the south due to the shipping and/or industrial emissions, the HOA concentrations exhibited a stronger enhancement at this hotspot than the rBC-rich factors.

Interpretation of the physical meaning of HOA and rBC-rich factors remains challenging. However, Dallmann et al. (2014) measured the rBC and OA concentrations in the emission plumes of on-road diesel trucks and gasoline cars as well as their ambient concentrations during the period with different diesel trucks traffic volume. Their results demonstrate that diesel truck emissions have a higher rBC content than that of gasoline vehicles, but the OA mass spectral features are similar between them. Therefore, the rBC-rich factor peaked one hour earlier than the HOA factor could be due to a relatively high contribution of heavy duty diesel trucks to the total traffic volume in the early morning in Singapore (Fwa et al., 1996).

The manuscript has been revised with such additional information to enhance the quality of our discussion.

Page 18, Lines, 6-15: Nevertheless, there are two major differences between the temporal variations of HOA$_{base-4}$ and rBC-rich $_{base-4}$ worth highlighting. Firstly, the diurnal profile of rBC-rich$_{base-4}$ peaked one hour earlier than that

of HOA$_{base-4}$ during morning rush hours. This could be due to the relatively high contribution of heavy-duty diesel trucks to the total traffic volume in the early morning of Singapore (Fwa et al., 1996). This speculation is well-aligned with an on-road investigation using SP-AMS that heavy-duty diesel trucks emit much higher rBC content than gasoline vehicles, but the OA mass spectral features in emission plumes are similar between them (Dallmann et al., 2014). Secondly, outside of the rush hours, both factors started increasing at ~12:00 LT and peaked at 17:00-18:00 LT that coincided with prevailing wind blown from the industrial and shipping region from the south direction. The NWR plots shows high concentrations of both HOA and rBC-rich factors at the hotspot observed from the south when the wind speed was greater than 8 m/s, but the HOA$_{base-4}$ concentrations exhibited a stronger enhancement at this hotspot than the rBC-rich$_{base-4}$ factors. Although most of the previous studies assigned their HOA and rBC-rich factors as local traffic-related OA (Carbone et al., 2019; Gentner et al., 2017; Malmborg et al., 2017), our results suggest that rBC-rich$_{base-4}$ and HOA$_{base-4}$ can be emitted from multiple combustion sources in this work. Nevertheless, the PMF$_{base}$ could not further resolve the contributions of traffic and shipping/industrial emissions to these two primary OA factors.

Fig. 8: should "Cother" be "C Other" in MO-OOA legend?

Response:

As the "Cother" in the original manuscript represented the metal signals, we have re-labeled "Cother" in all of our PMF analysis plots with "Metal".

Figure S6. "The NWR plots of factors identified by PMFbase-4." Would be helpful to point out the differences for these vs the plots in Fig. 8 (different WS resolutions, etc) in the caption.

Response:

The manuscript has been revised as shown below:

SI Page 15: Figure S7. The NWR plots of factors identified by PMF$_{base-4}$ with different wind speed ranges (< 3 m/s) and resolution than Fig. 8 (now Fig. 7).

Fig 11: Yellow metals bar color missing from legend

Response:

We have corrected and updated all the labels.

P29, L6: Should be Fig 9, not Fig 7

Response:

Figure number has been corrected.

P34, L4: Need "an" before "aged" or make "component" and verb made plural.

Response:

This grammatical error has been corrected as suggested.

P34, L13: hands shouldn't be plural

Response:

This grammatical error has been corrected as suggested.

**Reference**

Adam Reff, Shelly I. Eberly, and Prakash V. Bhave: Receptor Modeling of Ambient Particulate Matter Data Using Positive Matrix Factorization: Review of Existing Methods, J. Air Waste Manage. Assoc., https://doi.org/Journal of the Air & Waste Management Association, Vol. 57, No. 2, February 2007: pp. 146–154, 2012.

Cubison, M. J., Ortega, A. M., Hayes, P. L., Farmer, D. K., Day, D., Lechner, M. J., Brune, W. H., Apel, E., Diskin, G. S., Fisher, J. A., Fuelberg, H. E., Hecobian, A., Knapp, D. J., Mikoviny, T., Riemer, D., Sachse, G. W., Sessions, W., Weber, R. J., Weinheimer, A. J., Wisthaler, A., and Jimenez, J. L.: Effects of aging on organic aerosol from open biomass burning smoke in aircraft and laboratory studies, Atmos. Chem. Phys., 11, 12049–12064, https://doi.org/10.5194/acp-11-12049-2011, 2011.

Dallmann, T. R., Onasch, T. B., Kirchstetter, T. W., Worton, D. R., Fortner, E. C., Herndon, S. C., Wood, E. C., Franklin, J. P., Worsnop, D. R., Goldstein, A. H., and Harley, R. A.: Characterization of particulate matter emissions from on-road gasoline and diesel vehicles using a soot particle aerosol mass spectrometer, Atmos. Chem. Phys., 14, 7585–7599, https://doi.org/10.5194/acp-14-7585-2014, 2014.

Drinovec, L., Močnik, G., Zotter, P., Prévôt, A. S. H., Ruckstuhl, C., Coz, E., Rupakheti, M., Sciare, J., Müller, T., Wiedensohler, A., and Hansen, A. D. A.: The "dual-spot" Aethalometer: an improved measurement of aerosol black carbon with real-time loading compensation, Atmos. Meas. Tech., 8, 1965–1979, https://doi.org/10.5194/amt-8-1965-2015, 2015.

Fwa, T. F., Ang, B. W., and Goh, T. N.: Characteristics of truck traffic in Singapore, Journal of Advanced Transportation, 30, 25–46, https://doi.org/10.1002/atr.5670300204, 1996.

Lee, J. H. and Hopke, P. K.: Apportioning sources of PM2.5 in St. Louis, MO using speciation trends network data, Atmospheric Environment, 40, 360–377, https://doi.org/10.1016/j.atmosenv.2005.11.074, 2006.

Ma, M., Rivellini, L.-H., Cui, Y., Willis, M. D., Wilkie, R., Abbatt, J. P. D., Canagaratna, M. R., Wang, J., Ge, X., and Lee, A. K. Y.: Elemental analysis of oxygenated organic coating on black carbon particles using a soot-particle aerosol mass spectrometer, Atmos. Meas. Tech., 14, 2799–2812, https://doi.org/10.5194/amt-14-2799-2021, 2021.

Paatero, P. and Hopke, P. K.: Discarding or downweighting high-noise variables in factor analytic models, Analytica Chimica Acta, 490, 277–289, https://doi.org/10.1016/S0003-2670(02)01643-4, 2003.

Rivellini, L. H., Adam, M. G., Kasthuriarachchi, N., and Lee, A. K. Y.: Characterization of carbonaceous aerosols in Singapore: insight from black carbon fragments and trace metal ions detected by a soot particle aerosol mass spectrometer, Atmos. Chem. Phys., 20, 5977–5993, https://doi.org/10.5194/acp-20-5977-2020, 2020.

Willis, M. D., Healy, R. M., Riemer, N., West, M., Wang, J. M., Jeong, C.-H., Wenger, J. C., Evans, G. J., Abbatt, J. P. D., and Lee, A. K. Y.: Quantification of black carbon mixing state from traffic: implications for aerosol optical properties, Atmos. Chem. Phys., 16, 4693–4706, https://doi.org/10.5194/acp-16-4693-2016, 2016.

---

## Author Comment (AC2)

**Response to the reviewer comment 2:**

This study investigates the emission sources and atmospheric processing of refractory black carbon (rBC) in Singapore using a soot-particle aerosol mass spectrometer. By applying multiple positive matrix factorization (PMF) approaches, the analysis identified distinct sources of rBC. Incorporating metals in PMF distinguished industrial and shipping emissions linked to heavy metals, while integrating inorganic aerosols revealed aged biomass burning aerosols and interactions between secondary organic aerosols (SOA) and inorganic aerosol formation. These findings highlight the effectiveness of diverse PMF approaches in resolving complex urban aerosol sources and processes.

The paper is well-organized and makes a valuable contribution to the field of aerosol source and property investigations. Additionally, it offers significant insights into the potential of expanding the PMF method by incorporating metals to better explore and detail aerosol sources. These strengths make the paper suitable for publication in EGUsphere. However, there are several areas that require improvement for greater clarity. Therefore, I recommend the paper for publication pending major revisions.

Response: We thank for the positive comments from the reviewer over the significance of this work. We have added/updated some of the main texts to provide greater clarity. Our responses to specific comments are in blue color below:

General comments:

This paper provides valuable information on the properties of BC-coated aerosols. However, it is unclear whether the discussed properties represent typical features of BC-coated aerosols or are specific to aerosols in this region. For instance, the ion balance among inorganic components is significantly lower than 1, which the authors use to infer aerosol acidity. However, the provided values show limited variability and are much lower compared to NR-PM1. Is this discrepancy due to the specific location, the characteristics of BC-coated aerosols, or another factor? Other discussions on aerosol properties should address this issue to provide more clarity. Since the authors use ACSM to measure non-BC PM1, it would be beneficial to utilize that data to offer a broader perspective on the general characteristics of PM. Additionally, comparing ACSM data could help quantify the proportions of BC-coated versus non-coated aerosols, at least for non-refractory compounds. Such comparisons would enhance the readers' understanding of the results.

Response:

Thanks for the reviewer's input on this important aspect. In the original manuscript and SI, we compared the major chemical composition of $NR-PM_1$ (measured by the ACSM) and $NR-PM_{coating}$ (measured by the SP-AMS). The comparison provides quantitative results on the amount of $NR-PM_{coating}$ contributed to the speciated $NR-PM_1$ as mentioned by the reviewer. Based on the reviewer's suggestion, we have provided more details and/or extended the comparison between the chemical composition of $NR-PM_1$ and $NR-PM_{coating}$. Additional figure has been added to the SI (Fig. S5), showing the time series of the speciated $NR-PM_1$ and their fraction contribution to the total $NR-PM_1$. The specific comments regarding the aerosol acidity in $NR-PM_1$ have been addressed in responses to the specific comment later.

This work aims to advance our understanding of emissions and atmospheric processing of rBC-containing particles (including rBC cores and their $NR-PM_{coating,}$ but not total PM) in Singapore based on our improved PMF analysis results. The findings are important to understand air pollution from various fresh and aged combustion emissions, which are one of the main air pollutant sources in urban settings. We would like to emphasize that $NR-PM_1$ characterization is not the focus of this work, and reporting a full set of ACSM measurement results is a standalone study. Therefore, we would like to make a good balance between the depth of the discussion and the length of this manuscript. We have included more $NR-PM_1$ results in the main text and SI for improving clarification of some scientific arguments in the main text.

The following information has been added in the main text when we are comparing the major chemical composition of $NR-PM_1$ and $NR-PM_{coating}$ to guide the reader to the SI materials as shown below.

Page 11, Line 15 – Page 12, Line 3: The average mass concentration of $NR-PM_1$ measured by ToF-ACMS was $15.05 \pm 8.68$ μg/m$^3$ (Org = $7.06 \pm 4.64$ μg/m$^3$, $SO_4^{2-}$ = $6.16 \pm 4.29$ μg/m$^3$, $NO_3^-$ = $0.55 \pm 0.57$ μg/m$^3$, $NH_4^+$ = $1.44 \pm 1.03$ μg/m$^3$, and $Cl^-$ = $0.09 \pm 0.15$ μg/m$^3$). The time series correlation of SP-AMS and ToF-ACMS for Org, $SO_4^{2-}$, $NO_3^-$, $Cl^-$, and $NH_4^+$ are shown in Fig. S4b-f. Fig. S5 shows the time series and overall composition of $NR-PM_1$.

[Figure]

Figure S5. (a) Timeseries of NR-PM$_1$ measured by the ToF-ACSM and eBC measured by the aethalometer. (b) Mass fraction contribution of NR-PM$_1$ and eBC (left) and OA factors identified in NR-PM$_1$.

Moreover, regarding the PMF results, the significance of the sources could be better contextualized by providing the proportion of BC-containing particles relative to the total aerosol population. Without these additional details, the paper appears overly focused on methodology, making it challenging to translate the findings into meaningful air quality applications.

Response:

As mentioned above, this work aims to advance our understanding of emissions and atmospheric processing of rBC-containing particles (including rBC cores and their NR-PM$_{coating,}$ but not total PM) in Singapore based on our improved PMF analysis results. The findings are important to understand air pollution from various fresh and aged combustion emissions, which are one of the main air pollutant sources in urban settings. It is important to point out that rBC-containing particles can play a crucial role in radiative forcing for impacting climate, and co-emit and internally mix with toxic contaminants such as heavy metals and polycyclic hydrocarbon (PAH) from combustion processes and thus pose significant risks to human and ecosystem health.

The original manuscript has compared NR-PM$_1$ and NR-PM$_{coating.}$ In particular, based on the SP-AMS and ToF-ACSM measurements (i.e., NR-PM$_{coating}$ vs. NR-PM$_1$), ~16% of OA and less than 6% of each IA component were coated on rBC particles (Fig. S4). Following the response under the general comments above, we have also included the time series and pie chart of the bulk measurement by ACSM (Fig. S5), and a 4-factor PMF solution by ACSM (PMF$_{ACSM}$, Fig. S7). We have also included the comparison of PMF$_{ACSM}$ and PMF$_{all-8}$ in Section 3.6 of the revised manuscript. The details of ACSM data analysis have also been added to the SI.

Page 32 Line 7 – Page 33, Line 15:

**3.6 PMF comparison between ACSM and SP-AMS measurement**

[revised manuscript text omitted]

SI Pages 2

ToF-ACSM data analysis and PMF

The chemical characterization of NR-PM1 was obtained by a time-of-flight aerosol chemical speciation monitor (ToF-ACSM. Aerodyne Research Inc.) in 1-minute time resolution. A full description of the instrument can be found in Fröhlich et al. (2013). Similarly to the SP-AMS, it is composed of an aerodynamic lens that focuses the particle beam and directs it through three vacuum chambers, the last one being a detection chamber in which particles are vaporized by impaction on a Tungsten vaporizer heated at 600 °C and then ionized by electron impact (70 eV). Particles are then detected by the time of flight mass spectrometer. The ToF-ACSM data were processed by the Tofware (v3.2.4, Tofwerk AG, Thun, Switzerland). The ionization efficiency (IE) of nitrate and relative ionization efficiency (RIE) of ammonium and sulfate were calibrated before and after the measurement. The calculated and default RIE values of $NO_3^-$ (1.1), $SO_4^{2-}$ (0.46), $Cl^-$ (1.3), Org (1.4), and $NH_4^+$ (3.14) were used to quantify NR-PM1.

PMF analysis of OA measured by ACSM was performed by the Source Finder (SoFi, version 8) and ME-2 solver as described previously (Canonaco et al., 2013). Up to 10-factor solution was evaluated following the same approach as for SP-AMS PMF. A 4-factor solution was selected (referred as PMF$_{ACSM}$, Fig. S13). The four factors include HOA, cooking-related OA (COA), BBOA, and OOA. The subscript of ACSM indicates the PMF factors (i.e., HOA$_{ACSM}$) identified by ACSM measurements in the subsequent discussion. The mass spectral profiles, diurnal patterns, NWR plots, and PSCF plots of these four factors are reported in Fig. S13.

SI Page 21

[Figure]

Figure S13. The mass spectral profiles, diurnal patterns, NWR plots and 4-factor PMF solution of ACSM and their corresponding diurnal patterns.

When performing PMF with different methods, the authors downweight or multiply several inputs. It is important to clarify how these adjustments affect the results and how those compounds are treated in the final analysis.

Response:

A similar question was asked by reviewer 1 and answered with detailed weight-changing sensitivity analysis and corresponding effects on the PMF solutions. Briefly, the adjustments were made to create comparable S/N strengths between adjusted fragments (i.e., metals and IA fragments in this work) and OA fragments.

We add a few sentences to clarify the calculation in the results section.

The manuscript has been revised as shown below:

Page 8 Lines 23-25: The signal of $K^+$ was downweighed by 10 times, whereas $Rb^+$, $V^+$, and $Ni^+$ were upweighted by 3 times to generate comparable signal-to-noise ratio (S/N) strengths to OA and rBC fragments.

Page 8 Lines 28-29: Sulfate fragments' intensities were downweighed by 3 times due to their relatively strong S/N.

Regarding the inclusion of metals in PMF, although the authors use Hz signals rather than mass conversion, are there possibilities that different metal compounds evaporate at varying rates? Are all metals fully evaporated? While calibration may not have been necessary since the authors do not report metal masses, understanding the collection efficiency or ionization efficiency (IE) of metals is crucial to provide accurate ratios for certain sources.

Response:

We agree with the reviewer that an improved understanding of collection efficiency and ionization efficiency of metals can enhance the quality of measurements and subsequent analysis. A previous laboratory study has reported the relative ionization efficiency (RIE) of different metals, indicating that they likely have species-specific RIE values (Carbone et al., 2015). While we can apply their reported RIE values in our analysis, it remains unclear whether the RIE values of metals are instrument-dependent.

Therefore, the goal of our analysis is to include metal signals in PMF to improve the quality of our source apportionment analysis for OA and rBC (i.e., the metal signals used as a signature of specific emissions). However, the collection efficiency of metals remains unknown. The approach used in this work assumes that all detected metal species are internally mixed with rBC, thus having the same collection efficiency as rBC particles, as discussed in Section 2.4. It is important to note that a full understanding of collection efficiency and ionization efficiency requires systematic laboratory and field investigations, which are beyond the scope of this work and should be considered as a community effort.

The rBC mass fraction from SP-AMS is reported to be overestimated by 30%. Calibration for BC quantification is essential. Why does this overestimation occur, and is it consistent across other studies using SP-AMS measurements?

Response:

The SP-AMS and aethalometer use different measurement principles (mass-based vs. absorption-based) that could lead to intrinsic uncertainties. The aethalometer determines equivalent BC (eBC) based on aerosol absorption and assumption of mass absorption coefficients at specific wavelength. In this study, the SP-AMS is calibrated using Regal Black as outlined in our manuscript. However, it is important to point out that quantification of rBC using SP-AMS strongly depends on the degree of overlapping between LV and particle beam, which ultimately affects the RIE $_{rBC}$ values and collection efficiency of rBC particles (Willis et al., 2016). The coating thickness of rBC particles, which varies significantly within atmospheric BC particle population, can affect the particle beam width and subsequently the rBC quantification.

As aethalometer is more robust compared to SP-AMS for ambient BC measurement, the primary goal of our time series comparison is to demonstrate that rBC mass concentrations measured by the SP-AMS are reasonable and their temporal variations can be captured due to changes in BC source contribution. The detailed instrument comparison is out of the scope of this work. Nevertheless, it is worth mentioning that the percentage difference of BC concentrations between the two instruments observed in this work is comparable to those observed in previous field measurement (Nielsen et al., 2017; Rivellini et al., 2020; Xie et al., 2019) and laboratory study (Salo et al., 2024). More detail of the measurement techniques and uncertainties for BC particles has been previously reported (Lack et al., 2014).

The manuscript has been revised as shown below:

Page 11, Lines 11-15: The rBC concentrations measured by the SP-AMS was about 31% higher than the equivalent BC (eBC) concentrations measured by the aethalometer, a strong temporal correlation between the two measurements was observed, suggesting that the temporal variation of BC concentrations can be captured reasonably well by both instruments (R = 0.89, Fig. S4). Such strong correlation between the eBC and rBC concentrations was consistently observed at the same sampling location (Rivellini et al., 2020).

The authors mention that BBOA carries organonitrate, inferred from the NO/NO2 ratio. Does this mean the reported nitrate concentration is not entirely inorganic? Or has the organonitrate portion been separated out? If so, it is necessary to explain how inorganic nitrate and sulfate concentrations were derived, the potential for organonitrate inclusion, and, ultimately, how much organonitrate is present.

Response:

We did not provide evidence in detecting organo-sulfate in our discussion and hence our response will focus on organo-nitrate quantification. Due to the presence of organo-nitrate, the reported inorganic nitrate level represents the upper limit as there is no appropriate method to quantify organo-nitrate for AMS data measured by LV scheme. The calculation approach proposed by previous studies were based on the $NO^+/NO_2^+$ ratios of organo-nitrate species measured by the standard AMS measurements (i.e., TV vaporization scheme) (Day et al., 2022; Farmer et al., 2010). Given that the LV vaporization scheme used to detect rBC-containing particles in this work lead to different fragmentation patterns of OA compared to the TV vaporization scheme (Ma et al., 2021), it is questionable to apply the same calculation method to quantify organo-nitrate in rBC-containing particles in this work. For clarification, we have added a sentence to address the potential issue in the Method section as shown below:

Page 7 Lines 16-17: Given the potential contribution by organo-nitrates (see Section 3.5.2), the mass concentrations of inorganic nitrate likely represent their upper limit.

In several places, the authors discuss coating thickness without addressing size distribution. For instance, on Page 15, Lines 10–12, and in Section 3.3.1 on Lines 20, 24, and 27, comparisons are made to rBC, but it is unclear whether smaller or larger amounts of material compared to rBC explain the thin or thick coatings.

Response:

In this work we use relative term thin to refer to low Org/BC or $R_{BC}$ (e.g., <1) values that is commonly used in previous literatures of SP-AMS measurements, and thick coating was described with Org/BC values that are much greater than 1 (Cappa et al., 2012; Collier et al., 2018; Lee et al., 2019; Wang et al., 2017). We have added the coating thickness estimation using the sizing information obtained from the PToF measurements in SI as shown below. The main text has also been modified, pointing the reader to the SI when the coating thickness of each PMF factor are discussed.

SI Pages 2-3

Coating Thickness Estimation

Assuming the particles are spherical, the coating thickness of rBC-containing particles were estimated using the following equation. The relative density of OA and BC used in the estimation were 1.2 and 1.8 (El Mais et al., 2023; Fan et al., 2020), respectively.

$$t_{coat} = r_{part} - \sqrt[3]{\left(\frac{\rho_{coat}*r_{part}^3}{R*\rho_{core}+\rho_{coat}}\right)} \ [1]$$

where $t_{coat}$ is the coating thickness, $r_{part}$ is the particle radius, $\rho_{core}$ and $\rho_{coat}$ are the density of rBC and OA, respectively, $R$ is the OA-to-rBC ratio (Org/rBC).

The estimations of coating thickness at different particle sizes (100, 300 and 600 nm) are summarized in Table S2 as illustration. For example, an Org/BC ratio of 1 for a 100 nm particle will result in an estimated coating thickness of 13.2 nm, contributing to 26.3% of the overall particle diameter. The percentage contribution of coating to the overall particle diameter is a function of Org/rBC ratio at a given particle size. It is important to note that bare/fresh BC particles (i.e., low Org/rBC ratio), they are likely to be more fractal (Wang et al., 2021) which would lead to overestimation of the coating thickness.

Table S2. Coating thickness calculation for different particle sizes and Org/rBC ratios.

| Particle size (nm) | Org/rBC | Coating thickness (nm) | Coating thickness (%) |
|---|---|---|---|
| 100 | 0.1 | 2.3 | 4.6% |
| 100 | 1 | 13.2 | 26.3% |
| 100 | 10 | 30.2 | 60.3% |
| 300 | 0.1 | 6.8 | 4.6% |
| 300 | 1 | 39.5 | 26.3% |
| 300 | 10 | 90.5 | 60.3% |
| 600 | 0.1 | 13.7 | 4.6% |
| 600 | 1 | 79.0 | 26.3% |
| 600 | 10 | 180.9 | 60.3% |

Regarding size, in Section 3.2.4, the authors state that BC size increases with aging. However, generally, BC size does not increase; rather, coatings around BC grow with aging. The dual BC size distribution observed might suggest different sources. Alternatively, could the rBC signal include organic material, potentially explaining the observed aging effect?

Response:

The whole section of size distribution has been modified and moved to SI as suggested by another reviewer as no additional information could be obtained based on the UMR size distribution data. Rather, most of the important information has been provided by the PMF analysis.

We agree with the reviewer's comment that BC core size generally does not increase due to aging. Rather, the overall rBC-containing particle size may increase caused by the growth of coating on BC core during atmospheric aging. We have re-visit the discussion, but we did not include any statement about changes in BC core size due to aging. Our original manuscript also described the potential of multiple primary and secondary sources to rBC-containing particle materials (i.e., not only BC core itself). We have modified the related sentences to avoid potential confusion as much as possible as shown below:

SI pages 2:

Size distribution of rBC and associated coating materials

Figure S10 presents the size distributions of rBC and the major species of NR-PM$_{coating}$ (i.e., OA, NO$_3^-$, and SO$_4^{2-}$). rBC mass peaked at around 180 nm, and its concentration gradually decreased as a function of particle size. In this work, the majority of rBC-containing particle was emitted by local combination sources without substantial degree of atmospheric aging, which is supported by the results of source apportionment analysis (see Section 3.3-3.5) that ~81% of rBC was emitted from local traffic and industrial emissions. Bimodal distribution was clearly observed for OA, and the intensities of peaks at ~200 nm and 465 nm are comparable. This implies that OA in NR-PM$_{coating}$ were influenced significantly by both primary and secondary sources, which is consistent with the results of source apportionment analysis that ~41% and 49% of OA in NR-PM$_{coating}$ was due to local primary emissions and SOA, respectively. NO$_3^-$, and SO$_4^{2-}$ peaked between 500 and 600 nm, suggesting that most of these IA species were associated with more aged rBC-containing particles. Nevertheless, a weaker peak of NO$_3^-$, and SO$_4^{2-}$ at ~200 nm highlights that these IA might condense on freshly emitted rBC in a relatively short time scale when they were locally formed via photochemistry. The details of source apportionment results based on different PMF scenarios will be discussed in Section 3.3.

The size distributions of some fragment ions, including m/z 43, 44, 55, 57, and 60, are shown in Fig. S10b, and most of them also had bimodal distributions. Signals of m/z 43, 55, and 57 could be significantly contributed by both C$_x$H$_y^+$ and C$_x$H$_y$O$_z^+$ fragments from OA, but their relative contributions across different particle size cannot be resolved by the unit mass resolution (UMR) data. Although m/z 60 can be mainly contributed by C$_5^+$ from rBC

and $C_2H_4O_2^+$ from OA (i.e., the biomass burning tracer organic fragment), the size distribution of rBC, m/z 39 ($K^+$) and m/z 213 ($K_3SO_4^+$) (Fig. S10b-c) suggests that biomass burning-influenced rBC-containing particles observed in this work had relatively large particle size and $C_2H_4O_2^+$ is likely the key contributor to the m/z 60 signals within the larger size range.

Related to the above questions, are there any possibilities that rBC measurements contain organic contributions? Addressing this would help clarify the findings and improve the overall interpretation of the results in this manuscript.

Response:

It is possible that organics contribute to $C_1^+$ signals. In this work, $C_1^+$ mass contribution from rBC was constrained by the $C_1^+/C_3^+$ ratio obtained from our calibration standard (i.e., Regal Black), which is the best practice in the SP-AMS data analysis. $CO_2^+$ is another major fragment contributed by both rBC and OA. Note that $CO_2^+$ fragment from rBC is mainly due to the soot surface functionality and they are part of the refractory structure of soot particles, whereas $CO_2^+$ fragment from OA is mainly due to the presence of carboxylic acids formed via secondary processing in the atmosphere. All the above information has been included in the original manuscript (see section 3.3). In particular, the impacts of $CO_2^+$ and related fragments on calculating the $R_{BC}$ values of primary emission PMF factor were discussed.

Detail comments.

Section 3.1 page11: Presenting the ion balance of NR-PM$_{coating}$ measured by SP-AMS alongside the ion balance of bulk OA measured by ToF-ACSM could provide a clearer understanding of the differences in chemical composition.

Response:

Since we have more detail discussion in Section 3.2.3 for acidic sulfate formation, we have added the ionic balance of NR-PM$_1$ measured by ACSM when we are discussing the ionic balance of NR-PM$_{coating}$ measured by SP-AMS in Section 3.2.3. The average $NH_4^+_{meas}/NH_4^+_{pred \ ratios}$ calculated by the ACSM data of ~0.6 were observed. Our previous AMS measurement from May-June 2017 also reported an average value of ~0.6 for $NH_4^+_{meas}/NH_4^+_{pred}$ (Rivellini et al., 2020).

The discussion about the ionic balance in Section 3.2.1 has been removed. The related discussion has been modified in Section 3.2.3 as shown below:

Page 16, Lines 17-21: The average $NH_4^+_{meas}/NH_4^+_{pred}$ (Zhang et al., 2007) was ~0.47 most of the time, implying that acidic aerosol were locally formed via photooxidation that subsequently condensed on rBC particles and they could not be completely neutralized by gas-phase ammonia before they reached the site. The ionic balance of NR-PM$_1$ from ACSM measurements also shows that the average $NH_4^+_{meas}/NH_4^+_{pred}$ (~0.6) is consistent with previous standard AMS measurement in Singapore (Rivellini et al., 2020), further supporting the overall acidic aerosols.

We have added the $NH_4^+_{meas}/NH_4^+_{pred}$ calculated by the ACSM data in Fig. S4g. We agree with the comment that the $NH_4^+_{meas}/NH_4^+_{pred}$ in both NR-PM$_1$ and NR-PM$_{coating}$ do not change a lot over time, and we have the acidity related texts in Section 3.2.1.

[Figure]

Figure S4g. Diurnal pattern of $NH_4^+{}_{meas}/NH_4^+{}_{pred}$ calculated by ACSM data.

Section 3.1 page12: The observation that sulfate is the second most abundant component after BC in NR-PMcoating and it is supported by previous literature. However, this finding could be further substantiated by including the time series and a pie chart of bulk OA composition, which would provide additional context and strengthen the interpretation of the results.

Response:

Since the focus of this paper is to investigate rBC and NR-PM$_{coating}$, we have decided to include the time series and chemical composition pie chart of ACSM measurements in SI, balancing the depth of the discussion and the length of this manuscript. Based on the ACSM results shown in Fig. S5, sulfate is the second largest component after OA in NR-PM$_1$ component, which is consistent to our previous study at the same location as described in the original manuscript. The statement regarding the observation that rBC particles were not the major condensation sink for the secondary IA components (i.e., less than 6%, Fig. S4) remained unchanged.

Section 3.2.2 page15: To explain the low values of Org/BC, O:C, and OSc observed during 8:00–9:00 LT and 20:00–21:00 LT as being caused by POA, it would be necessary to also analyze the diurnal patterns of vehicle emissions or SOA based on PMF results. While, previous studies reported that POA (particularly from traffic) tends to increase during these time periods, this cannot definitively confirm that the observed values are due to POA in this study. Still the interpretation remains uncertain. Therefore, it is recommended to present the explanation related to POA as a possibility or to integrate it into the discussion of PMF results at Section 3.3.2, which provide a more comprehensive context for such interpretations.

Response:

We agree with the reviewer that we may over-interpret some of the results based on the O:C ratio alone in this section. As the PMF results reported in the later sections provide evidence to support our argument, we have revised the manuscript as follows.

While we mentioned the possibility of POA emissions during traffic rush hours in this section, we have moved the key argument about the characteristics of traffic emissions (i.e., HOA) to the later section with the support from PMF results as the reviewer suggested.

Section 3.2.2 page16: To interpret the increase in $fC_2H_4O_2^+$ between 2:00 and 8:00 LT as being influenced by oxidized biomass burning emissions, it is important to consider previous studies that suggest the $fC_2H_4O_2^+$ signal decreases as aging progresses. This suggests that biomass burning was active during this period and, separately, secondary production and aging of other OA components may have contributed to the observed increase in $OS_c$.

Response:

Cubison et al. (2011) is one of the important studies in the literature to show the effects of atmospheric aging on $f_{C_2H_4O_2^+}$ values in biomass burning emissions. We have revised the manuscript by including the $f_{C_2H_4O_2^+}$ background level (e.g. for non-biomass burning OA or heavily aged biomass burning OA) based on the observations from Cubison et al. (2011). In this work, the background level was adjusted based on our previous study due to the changes in fragmentation pattern using LV scheme of SP-AMS (Ma et al., 2021). A dashed line indicating potential $f_{C_2H_4O_2^+}$ background level has been added to Fig. 6d for visualizing the results. Noting that the $f_{C_2H_4O_2^+}$ of total OA was generally below the background level but the elevated levels were observed between 2:00 and 8:00 LT, suggesting the potential influences of fresh/aged biomass burning emissions during the period. Since this section aims to provide an overview of OA characteristics, the potential contribution of fresh and aged biomass burning emissions will be further discussed based on the PMF analysis in Section 3.3. The related discussion has been revised as shown below:

Page 15, Line 19 – Page 16, Line 9: To investigate the possible biomass burning influences on OA coatings, a mass contribution of a biomass burning tracer fragment to the total organic coating ($f_{C_2H_4O_2^+}$) was determined as shown in Fig. 6d. Cubison et al. (2011) reported that $f_{C_2H_4O_2^+} > 0.3\%$ obtained from standard AMS measurements can be the indication of biomass burning influences. Ma et al. (2021) showed that $f_{C_2H_4O_2^+}$ can be enhanced by a factor of ~2.45 on average for oxygenated organic coating and 2.33 for levoglucosan vaporized by the LV scheme of SP-AMS. Therefore, the background level of $f_{C_2H_4O_2^+}$ was re-calculated using the correction factor of 2.33 as shown by the dashed horizontal line in Fig. 6d. Although the $f_{C_2H_4O_2^+}$ values of total OA were close to the background level, the higher values of $f_{C_2H_4O_2^+}$ and $OS_c$ of OA coating materials between 2:00 and 8:00 LT suggests potential influences of biomass burning emissions on the chemical characteristics of rBC-containing particles during the night. The potential contribution of fresh and aged biomass burning emissions will be further discussed based on the PMF analysis (see Section 3.3).

Section 3.4.1 page25: The use of the $V^+/Ni^+$ ratio as evidence for shipping emissions is reasonable and aligns with previous studies. However, the classification of the industrial-related factor as IOA requires additional supporting data. Specifically, further analysis of other metal ratios could provide stronger evidence to validate its association with industrial emissions. Including these ratios would enhance the reliability of the interpretation and clearly differentiate industrial contributions from other sources.

Response:

We agree with the reviewer that including more metal and metal ratios in the PMF analysis can enhance the quality of source apportionment analysis. However, we have included all the metals that were above the detection limit in this study.

---

## Author Response (AR2)

Public justification (visible to the public if the article is accepted and published):

This is clearly a very substantial and novel piece of work and while the authors have gone to efforts to address the comments of the reviewers, I feel that much of the argument made in the rebuttal is not adequately reflected in the revised document. In general, in spite of the reviewers highlighting the issue, the paper does risk being seen as being 'too technical' or 'too local' to be considered as a research article in ACP. So I would like to extend the opportunity to the reviewers to further revise the article. However because these modifications do not concern the scientific content, I have decided that a further round of peer review is not likely to be necessary.

Response: We thank for the positive feedback from the editor on the significance of this work. Our responses to specific comments are in blue color below:

1. The discussion and conclusions need to more clearly show the wider implications rather than just the technical developments and the observations. In particular, the authors must show that the aerosols observed aren't specific to Singapore and have relevance for the wider atmosphere. While additional articles describing air pollution in Singapore are referred to in the introduction, the wider context needs to come through in the discussion and conclusions, so I would suggest more attention being paid to comparing observations to these previous works. Furthermore, the introduction and conclusions should also describe how typical Singaporean air quality is compared to other locations in the region and following this, a commentary on other areas these observations may be applicable to. These may seem trivial to someone already familiar with the location and the associated literature, but are important bits of contextual information for ACP's general readership.

Response: The abstract, introduction and conclusion have been modified based on editor's suggestions. In particular, the general implications of individual PMF factors identified in this work have been discussed in detail in the conclusion.

The abstract has been revised to reflect the updated main text:

Page 2 lines 4-8: Positive matrix factorization (PMF) analysis of rBC and organic aerosols (OA) ($PMF_{base}$) identified two traffic factors with differences in rBC content, coating thickness and diurnal pattern, which could potentially help differentiate gasoline and diesel vehicular emissions. Additionally, two secondary OA (SOA) factors influenced by local chemistry and/or regional transport (less-oxidized oxygenated OA (LO-OOA) and more-oxidized OA (MO-OOA)) were identified. Including metals in the PMF ($PMF_{metal}$) improved the quality of source apportionment significantly.

Page 2 lines 11-13: Although the aged BBOA component was highly oxidized, its strong association with $K_3SO_4^+$ distinguished it from other background MO-OOA, which generally lacked distinctive OA signatures.

The manuscript has been revised as shown below:

Page 4 lines 15-29: Singapore is a highly developed coastal urban environment located at the southern tip of the Malay Peninsula in Southeast Asia. In additional to typical local emissions from vehicular traffic and cooking , Singapore hosts the world's second busiest shipping port (World Shipping Council, 2024) and fifth largest oil refining capacity (Lau et al., 2021), both of which are potentially significant anthropogenic sources of combustion-related PM and toxic pollutants to impact the local air quality. Moreover, air quality and concentrations of atmospheric PM in Singapore can be significantly influenced by SOA and transboundary haze caused by biomass burning emissions transported from surrounding regions in Southeast Asia (Budisulistiorini et al., 2018; Heil and Goldammer, 2001; Kuwata, 2024). Given the unique capability of the SP-AMS to simultaneously detect rBC particles and their associated coatings, this study explores multiple PMF scenarios based on SP-AMS measurements, iteratively incorporating rBC, OA, inorganic aerosols (IA), and metals into the input. This step-by-step PMF interpretation approach underscores the value of including all measured chemical species in the analysis to better resolve local primary emissions and the secondary transformation processes of rBC-containing particles in Singapore. The improved characterization presented in this study provides important insights into how industrial emissions may introduce toxic heavy metals and influence local SOA formation in a complex urban environment. It also sheds light on how regional biomass burning may affect background levels of rBC-containing particles in Southeast Asia's urban atmosphere.

[revised manuscript text omitted]

2. On a technical note, while the downweighting is described a little better in the text, the justification given in the rebuttal should also be included in the revised text regarding the 'strength' of the variables and what precedent there is in the literature for using the method. This is an essential part of the novel aspect of the methodology, so it must be properly documented.

Response:

We agree with the comment and have added Fig. R1 to SI and included the justification in the main text. SI figure numbers were also updated accordingly due to the new SI figure.

The manuscript has been revised as shown below:

Pages 8-9, lines 23-2: The signals of $K^+$, $Rb^+$, $V^+$, and $Ni^+$ were either downweighted or upweighted to produce SNR comparable to those of typical OA and rBC fragments. For instance, strong fragments such as $C_2H_3O^+$ and $C_3^+$ exhibit SNRs of ~25 and ~21, respectively. To bring $K^+$ into a comparable range, its signal was downweighted by a factor of 10, resulting in an SNR of ~13. In contrast, the signals of $Rb^+$, $V^+$, and $Ni^+$ were each upweighted by a factor of 3 to achieve SNRs of ~6. Figure S1 demonstrates how adjusting SNR of $K^+$ improves the performance of PMF as an example. Previous studies of SP-AMS measurements have also used a similar approach to integrate metal-related ions into PMF analysis (Bibi et al., 2021; Carbone et al., 2015; Rivellini et al., 2020). To further investigate the nature and formation of OA, IA-related fragments from sulfate, nitrate, and chloride were included in the PMF analysis. This PMF analysis included all the measured aerosol components (i.e., OA + rBC + metal + IA) (denoted as $PMF_{all-n}$ hereafter). Sulfate ions that exhibited SNRs between 10 and 30 were downweighted by a factor of 3. Nitrate ions and $K_3SO_4^+$ had SNR values in the range of 5 to 15 that do not require adjustment. Noting that downweight and upweight signals of AMS data for PMF analysis has been more systematically investigated by previous studies to improve the separation of factors (Adam Reff et al., 2012; Lee and Hopke, 2006; Paatero and Hopke, 2003).

SI Page 9

[Figure]

Figure S1. The comparison of 7-factor solutions using default S/N with $K^+$ dominated factor (i.e., poor separation, a), adjusted S/N in $PMF_{metal-7}$ (b), and over-adjusted S/N showing over-separation (i.e., S/Ns < 0.2, c).